# Parental socioeconomic composition of birth cohorts changed during the COVID-19 pandemic

Moritz Oberndorfer [1,2,3] ✉, Juha Luukkonen [1,2], Hanna Remes [1,2], Thomas Waldhör[4], Lizbeth Burgos-Ochoa [5,6], Márta K. Radó[7], Jasper V. Been [6,8], Olof Mpumwire Östergren[9], Peter Fallesen [10,11], Alicia Montgomerie [12,13], Rhiannon Megan Pilkington[12,13], John Lynch [12,13,14], Enny S. Paixao Cruz[15,16], Ila R. Falcão[15] & Pekka T. Martikainen [1,2,9,17]

The COVID-19 pandemic offers opportunities to study effects of in-utero and early life exposure to environmental changes. However, inferences from such studies may be flawed if the pandemic has changed the socioeconomic composition of parents. Analysing over 77.9 million live births from 15 countries, we estimate changes in the socioeconomic composition of the cohort born between December 2020 and December 2021 using interrupted time series analysis. We find that, compared with their counterfactual compositions, the December 2020-December 2021 birth cohort has a higher proportion of babies born to socioeconomically advantaged parents in Austria, England, Finland, the Netherlands, Scotland, Spain, Wales, and the United States while we observe the opposite change for Brazil, Colombia, Ecuador, and Mexico. These changes in cohort composition may cause between-cohort differences in life course outcomes that are influenced by parental socioeconomic circumstances even if early life exposure to the pandemic had no direct effect on this birth cohort.

The Coronavirus Disease (COVID)−19 pandemic has created "a natural experiment of unprecedented proportions"[1]. As such, it is widely used as an exposure in a growing number of natural experimental study designs to uncover causal relationships of core interest in a range of scientific disciplines[1].

These studies seek to use the unanticipated and sudden occurrence of a pandemic-induced temporary change in the environment humans live in as exogenous exposure. However, the most severe caveat of such study designs is that the COVID-19 pandemic might have simultaneously changed human behaviour, making it difficult to isolate the effect of a single pandemic-related exposure on an outcome[2].

Research interested in the effects of in-utero and early life exposures has already used the COVID-19 pandemic as a natural experiment to study pregnancy and birth outcomes[3–5]. This research is often

motivated by or interpreted through the 'foetal origins hypothesis'[6,7] which proposes that a foetus adapts to the maternal environment during gestation[6–8] and that so-called foetal programming may have long-term effects on health, developmental, and socioeconomic outcomes throughout the life course[6]. As the cohorts born and conceived during the COVID-19 pandemic age, we expect this literature to expand rapidly across many outcomes like educational attainment, income, mental health, and mortality in later stages of the life course.

However, differences in outcomes between cohorts born and conceived before, during, or after the COVID-19 pandemic can only be causally attributed to in-utero or early life exposure to the pandemic if the exposed and unexposed cohorts are exchangeable[9]. Intuitively, because the pandemic was an unanticipated population-wide event, it might seem safe to assume that there is no confounding variable that is associated with an outcome and caused in-utero or early life exposure

to the pandemic. Numerous already published epidemiological studies on how the pandemic affected birth and pregnancy outcomes implicitly assume exposed and unexposed cohorts to be exchangeable[10–12].

While not obvious, we believe this assumption is likely violated when using the COVID-19 pandemic as an in-utero or early life exposure. For babies born and conceived during the pandemic, there are two main selection mechanisms which can violate the exchangeability assumption and potentially bias effect estimates of in-utero and early life exposure to the pandemic on life course outcomes.

First, for babies conceived before the pandemic but exposed to the pandemic in-utero, this bias can occur due to selection mechanisms while in utero[8,13] (e.g., through pregnancy loss, abortion, miscarriages, stillbirths) – also known as live birth bias[14] in perinatal epidemiology. The second mechanism – selection into conceptions – is the focus of this paper. For babies conceived during the pandemic, the exchangeability assumption is violated if the composition of conceptions and, consequently, the composition of live births with respect to parental characteristics that are associated with the outcomes of interest changed during the pandemic. We believe that such compositional changes in parental socioeconomic characteristics are especially likely in the context of the unequal impacts of the COVID-19 pandemic[15–18]. There are many plausible hypotheses for why fertility responses to the pandemic might have differed across socioeconomic or age groups[19] and empirical evidence has also shown differing fertility responses[20–28]. For example, a study using Norwegian register-based data has concluded that an increase in fertility during the pandemic was driven by women aged 28-35, women who already have children, and women with tertiary educational attainment[22]. For Spain, Cozzani et al. have found a pandemic-induced decline in fertility, especially among women without tertiary educational attainment[21].

Socioeconomically advantaged groups were expected to adjust their fertility during the pandemic to positive changes in their work-life balance (e.g., working from home and reduced travel)[20]. At the same time, restricted access to assisted reproductive technology[21] (e.g. fertility clinics were closed during the lockdown) may have been a barrier for fertility – especially for those at higher reproductive ages. On the other hand, the pandemic may have led socioeconomically disadvantaged families to postpone pregnancies in the face of increased economic uncertainty and income losses[19,20,22]. Further, the fertility response of socioeconomically disadvantaged groups may be more sensitive to their access to contraception[29]. Importantly, explanations for pandemic-induced change in parental socioeconomic composition of births must also pay attention to context-specific reasons for pre-pandemic socioeconomic differences in fertility[30].

Based on previous literature in health and social sciences, we believe that the existence of this bias is highly plausible for two reasons. First, emerging evidence from demographic research suggests that the effect of the COVID-19 pandemic on fertility was dependent on parental socioeconomic circumstances at birth in Scotland[20], Spain[21], Norway[22], Iceland[31], Sweden[23], the United States[24,25], Colombia[26], Brazil[26], Mexico[27], and Australia[28,32]. Second, because there is robust evidence that parental socioeconomic circumstances are strongly associated with health, developmental, and socioeconomic outcomes throughout the life course[33,34], it is plausible that sudden changes in the parental socioeconomic composition of a birth cohort can cause population-level differences in outcomes between babies conceived during the pandemic and earlier and later cohorts.

Demographic research focused on fertility changes during COVID-19 pandemic[22,23,26,31,35,36] is not necessarily concerned with the consequences of changed fertility for life course outcomes of babies born and conceived during the pandemic. Conversely, medical research which focuses on the COVID-19 pandemic's potential effects on perinatal outcomes like preterm birth, birth weight, or stillbirths rarely discusses how selection in-utero and/or selection into conception may affect their conclusions regarding found improved or worsened perinatal outcomes[10–12]. Yet, compositional changes through selection in-utero and/or selective conception could explain changes in perinatal outcomes during the pandemic. For example, Catalano et al. argue that, for the United States, the decline in preterm births for babies conceived before but born during the pandemic was caused by selection in-utero[37]. Similarly, Cozzani et al. argue that the positive changes in preterm births among babies conceived during the pandemic are caused by selective conception in Spain[21].

Not accounting for compositional changes when studying in-utero exposure to a pandemic on life course outcomes has led to erroneous conclusions in the social and health sciences before. Only in 2022, it has been shown that, in the United States, war-induced changes in the socioeconomic composition of parents during the 1918 influenza pandemic explain why the 1919 birth cohort had lower adult socioeconomic status than earlier and later birth cohorts[38]. Previously, this difference was thought to be caused by in-utero exposure to the 1918 influenza pandemic[6].

Thus, if the COVID-19 pandemic has led to sudden changes in the parental socioeconomic composition of the cohort born and conceived during the pandemic, research comparing (persons from) this birth cohort to other cohorts studying trends in life course outcomes needs to take this compositional change into consideration.

In this study, we analyse population-wide administrative data on 77.9 million live births from 15 countries covering the period 2015-2021 and interrupted time series analysis to answer if, and to what extent, the cohorts of live births conceived during the COVID-19 pandemic (born between December 2020 and December 2021) have a different parental socioeconomic composition than expected had there been no pandemic. We thereby replicate and expand existing country-specific studies on socioeconomic inequalities in fertility changes during the pandemic for 8 of 15 countries (Brazil[26], Colombia[26], Mexico[27], Scotland[20], South Australia[28], Spain[21], Sweden[23], the United States[24,25]) and add evidence for the other 7 included countries (Austria, Denmark, Ecuador, England, Finland, Netherlands, Wales). Apart from this, our comparative study makes two contributions. First, we bridge the literature in demography on heterogenous fertility responses to the pandemic and the literature in epidemiology on the pandemic's effect on perinatal outcomes by providing comparable evidence on compositional changes that can cause between-cohort differences in perinatal outcomes and other life course outcomes. We thereby aim to spread awareness about potential long-term implications of changes in the socioeconomic composition of babies conceived during the pandemic beyond the sphere of fertility research. Second, our comparative design is not limited by variation in the specific research questions and methodological approaches as present across the existing country-specific literature. This enables us to more convincingly learn about generalisable mechanisms based on similarities and differences in the country-specific findings. Our selection of countries is based on differences in pre-pandemic social inequalities and differences in fertility trends, welfare regimes, pandemic mitigation policy, pandemic experiences, and data availability. For Denmark, Sweden, and South Australia, we find little to no evidence for pandemic-induced changes in the parental socioeconomic composition of live births. For seven out of nine included European countries (Austria, England, Finland, the Netherlands, Scotland, Spain, Wales) and the United States, we find that babies conceived during the pandemic had a more socioeconomically advantaged parental composition than expected had pre-pandemic trends continued. For all Latin American countries (Brazil, Colombia, Ecuador, and Mexico) included in our analysis, we find the opposite: the composition of live births conceived during the pandemic changed towards more socioeconomically disadvantaged groups. The percentage point differences in the proportion of live births born to socioeconomically advantaged groups were between −1% and +1%, except for Spain ( + 2.5%) and Ecuador (−3%). Similarly, the percentage point differences in the proportion of babies born to

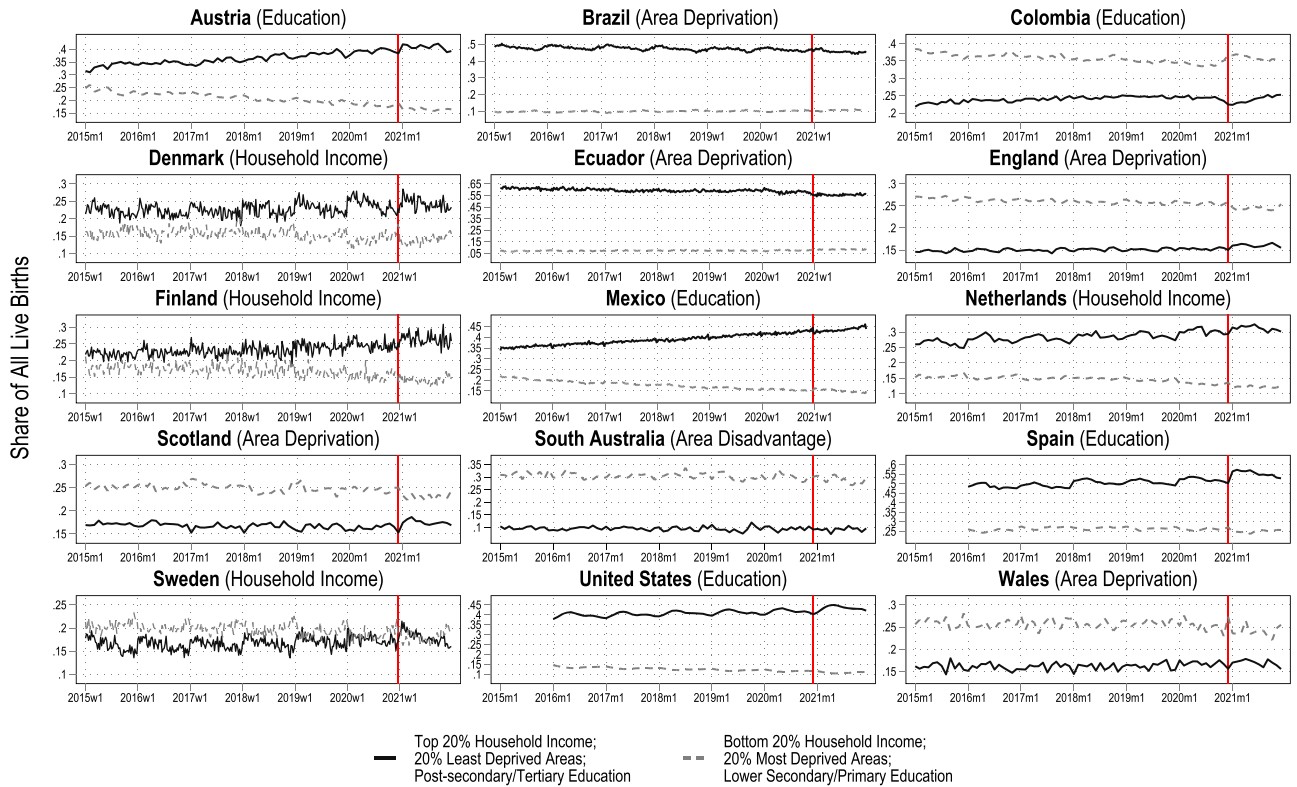

**Fig. 1 | The proportion of live births by most (dashed grey lines) and least socioeconomically disadvantaged (solid black lines) groups.** The solid vertical line marks the start of the exposed period. The indicator of socioeconomic circumstances used for each county is mentioned in parentheses next to the country name. Time series are weekly for Brazil, Denmark, Ecuador, Finland, Mexico, and Sweden, and monthly for the other countries. Note that, for Mexico, due to data availability, the black solid line presents the share of live births by women with upper secondary educational attainment and the grey dashed line the share by women with elementary educational attainment. Also note that the y-axes are different which impairs visual comparability of changes between countries. For Spain, only births with maternal age over 25 are included (see methods and suppl. material).

socioeconomically disadvantaged groups were between −1% and +1%, except for Colombia (+2.6%). We conclude that these changes in socioeconomic composition may cause between-cohort differences in life course outcomes that are affected by the socioeconomic position of parents even if in-utero or early life exposure to the pandemic had no direct effect on these outcomes. Cross-country similarities and differences in our results suggest that changes in a birth cohort's socioeconomic composition in response to macro-level shocks depend on both policy responses and on how socioeconomic position relates to agency in fertility decisions in different societies.

## Results

The analysed data covered 77.95 million live births across 15 countries born between 2015-2021, out of which over 10.9 million live births were conceived during the COVID-19 pandemic. We first give a descriptive overview of trends in the proportion of live births born to the socioeconomically most disadvantaged and advantaged groups in each country in Fig. 1. Next, we show the estimated observed and counterfactual trends in the number of births in these groups in Figs. 2 and 3 to present our analytical approach and the changes in the number of births that underlie the estimated counterfactual socioeconomic composition of birth cohorts. Finally, we present our main results, the percentage-point differences between the observed and the counterfactual socioeconomic compositions, in Fig. 4. The most important results visualised in Figs. 1–4 are then summarised in Supplementary Table 1.

In Fig. 1, we visualised the observed weekly/monthly proportions of live births by women living in households placed the highest 20% of

the country-specific equivalised household income distribution (Denmark, Finland, Netherlands, Sweden), living in the 20% least deprived areas of their country (Brazil, Ecuador, England, Scotland, South Australia, Wales), or by mothers who completed post-secondary or tertiary educational attainment at birth (Austria, Colombia, Mexico, Spain, the United States) in black solid lines – henceforth called the socioeconomically advantaged groups. Displayed by grey dashed lines, Fig. 1 also shows the observed weekly/monthly proportions of live births by mothers living in households placed the poorest 20% of the country-specific household income distribution (Denmark, Finland, Netherlands, Sweden), living in the 20% most deprived areas of their country (Brazil, Ecuador, England, Scotland, South Australia, Wales), or by mothers who had primary or lower secondary educational attainment at birth (Austria, Colombia, Mexico, Spain, United States) – henceforth called the socioeconomically disadvantaged groups. The vertical line depicts the start of the exposed period (51st week of 2020 for Brazil, Denmark, Ecuador, Finland, and Mexico; and December 2020 for Austria, Colombia, England, the Netherlands, Scotland, South Australia, Spain, the United States, and Wales).

In Austria, Denmark, England, Finland, the Netherlands, Scotland, Spain, Scotland, and Sweden, the proportion of live births held by the socioeconomically advantaged groups climbed to its highest values at the start of the exposed period. In contrast, the proportions of the socioeconomically most disadvantaged groups increased in Colombia, Ecuador, and Mexico.

We visualised the observed numbers of weekly or monthly live births (dots), the estimated numbers (grey solid line), the deseasonalised estimates (solid black lines), and the counterfactual numbers

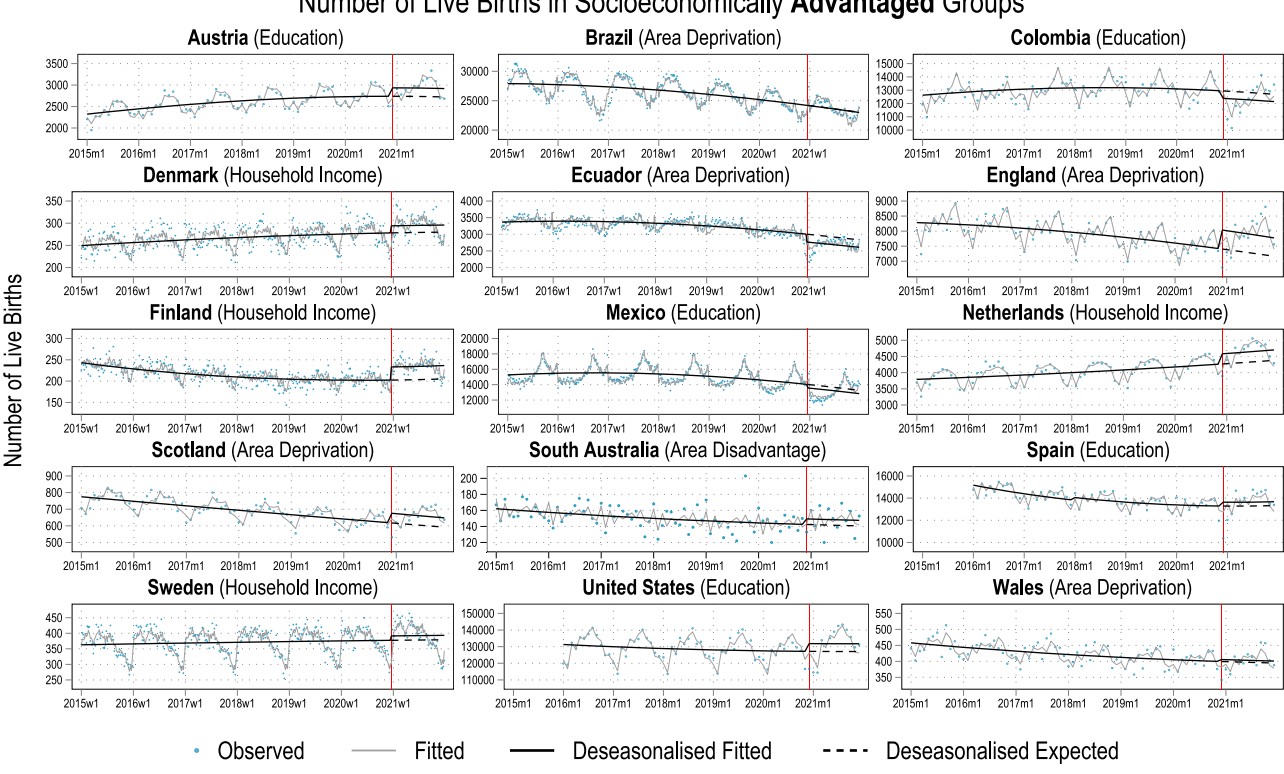

Fig. 2 | **Number of live births 2015-2021 among socioeconomically advantaged population groups in all included countries.** Blue dots indicate the observed number of live births. The solid vertical line marks the start of the exposed period. Grey solid lines indicate the number of live births estimated by group-specific Poisson regression modes on the full time series including an indicator variable for the exposed period to estimate the average effect of the COVID-19 pandemic over the entire period; a linear and a quadratic term for week/month of live birth to capture potential non-linearities in the secular time trends; week/month of the year fixed effects to account for seasonality; an indicator variable for August 2016 to December 2016 for Brazil, Ecuador, Colombia, and Mexico to account for the Zika Virus epidemic; an indicator variable for 2016 and 2017 for Spain to account for changes in data collection. Black solid lines show the deseasonalised trends estimated from these models, ignoring any level changes due to the Zika Virus epidemic. Black dashed lines show the estimated deseasonalised expected (counterfactual) number of live births had the COVID-19 pandemic never happened and pre-pandemic trends continued instead. Socioeconomic indicators used are shown in parentheses on top of each panel. For Spain, only births with maternal age over 25 are included (see methods and suppl. material).

(dashed black lines) born to the most socioeconomically advantaged groups in Fig. 2 and these respective numbers for those born to the most socioeconomically disadvantaged groups in Fig. 3. In Supplementary Table 1, we compare the socioeconomic compositions of the observed cohorts conceived during the COVID-19 pandemic with their respective counterfactual compositions. Due to space limitations, uncertainty estimates are provided visually in Fig. 4 and in the supplementary Tables 2–16 at the end of each country profile.

In Austria, Denmark, England, Finland, the Netherlands, Scotland, Spain, and the United States and the average number of weekly/monthly live births in the socioeconomically advantaged groups increased visibly between December 2020 and December 2021 (Fig. 2). Relative to the counterfactual number of live births during this entire period, socioeconomically advantaged groups had 7.2% [95%CI: 5.2%; 9.2%] (Austria), 5.7% [95%CI: 2.9%; 8.7%] (Denmark), 8.5% [95%CI: 7.3%; 9.7%] (England), 15.3% [95%CI: 11.7%; 19%] (Finland), 7.4% [95%CI: 5.8%; 9.0%] (Netherlands), 9.4% [95%CI: 5.5%; 13.7%] (Scotland), 2.6% [95%CI: 1.6%; 3.6%] (Spain), and 3.7% [95%CI: 3.4%; 4.0%] (United States) more live births than expected based pre-pandemic trends (Fig. 2, Supplementary Table 1, Supplementary Fig. 2). In these countries, the socioeconomically disadvantaged groups either experienced small increases in the number of live births as well (Austria: 3.3% [95%CI: 0.6%; 6.0%], Netherlands: 3.3% [95%CI: 1.1%; 5.6%]) or small decreases (England: -0.4% [95%CI: -1.2%; 0.5%], Scotland: -1.3% [95%CI: -4.3%; 1.9%], Spain: -2.3% [95%CI: -3.6%; -1%], United States: -2.4% [95%CI: -3%;

-1.9%]) compared to their counterfactual number of live births (Fig. 3, Supplementary Table 1, Supplementary Fig. 2). Denmark and Finland are exceptions, where the lowest household income group (Denmark: 11.8% [95%CI: 8.1%; 15.8%], Finland: 10% [95%CI: 5.9%; 14.5%]) as well as all other income groups showed substantial increases in birth cohort size between 5.5% and 11.8% (Supplementary Table 1, Supplementary Fig. 2).

In Colombia, Ecuador, and Mexico, the observed numbers of live births among the socioeconomically advantaged groups were lower than the counterfactual numbers (Fig. 2). In Brazil and Wales, the observed and counterfactual trends differed only negligibly for socioeconomically advantaged groups. In Brazil, Colombia, and Ecuador, the number of live births born to the socioeconomically most disadvantaged was higher than their counterfactual between December 2020 and December 2021 (Fig. 3). Relative to the counterfactual number of births, 5% [95%CI: 4.3%; 5.6%] more babies were born to women living in the most deprived municipalities of Brazil, 5.7% [95%CI: 5.0%; 6.5%] more babies were born to women with primary or lower secondary educational attainment in Colombia, and 8.7% [95%CI: 6.2%; 11.3%] more babies were born to women living in the most deprived cantons of Ecuador (Supplementary Table 1, Supplementary Fig. 2).

The size of these differences was dominated by the initial change in the number of births and the consequent recovery during 2021. Among socioeconomically advantaged and disadvantaged groups in Austria, Brazil, Denmark, England, Finland, the Netherlands, and

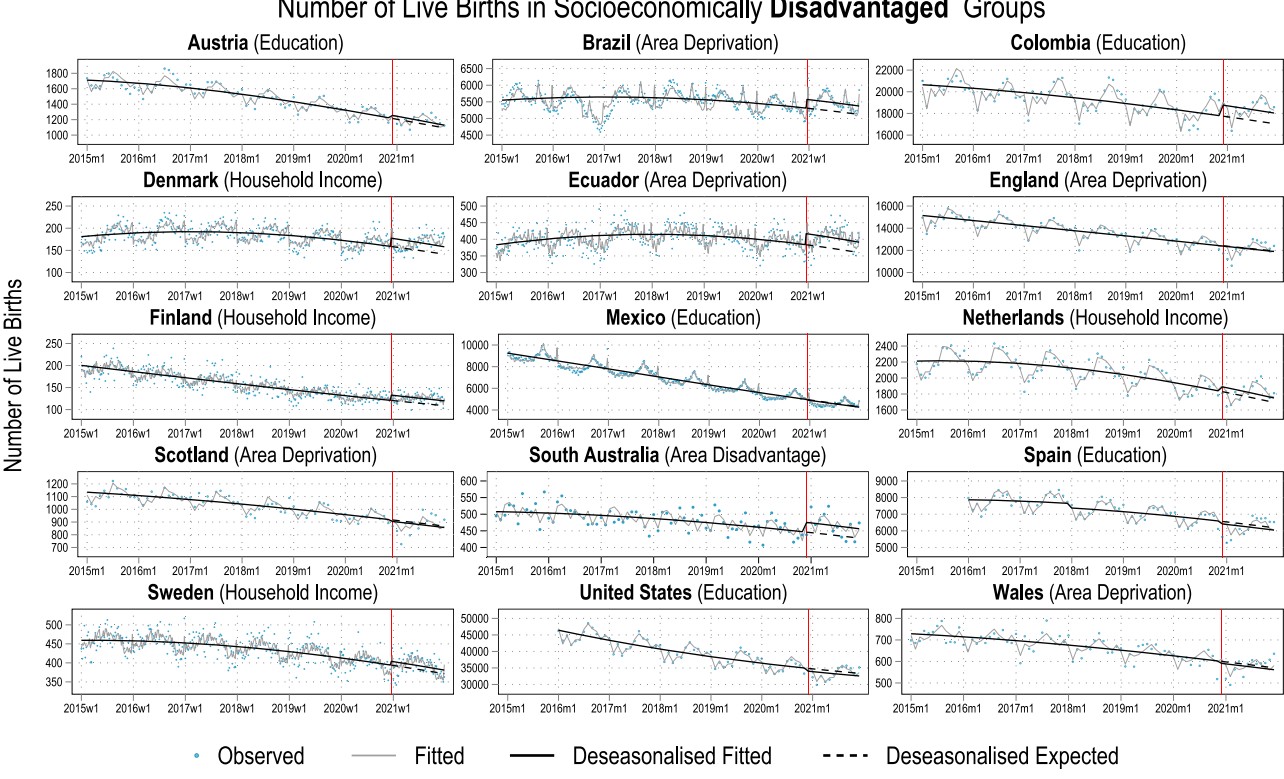

**Fig. 3 | Number of live births 2015-2021 among socioeconomically dis-advantaged population groups in all included countries.** Blue dots indicate the observed number of live births. The solid vertical line marks the start of the exposed period. Grey solid lines indicate the number of live births estimated by group-specific Poisson regression modes on the full time series including an indicator variable for the exposed period to estimate the average effect of the COVID-19 pandemic over the entire period; a linear and a quadratic term for week/month of live birth to capture potential non-linearities in the secular time trends; week/month of the year fixed effects to account for seasonality; an indicator variable for August 2016 to December 2016 for Brazil, Ecuador, Colombia, and Mexico to account for the Zika Virus epidemic; an indicator variable for 2016 and 2017 for Spain to account for changes in data collection. Black solid lines show the desea-sonalised trends estimated from these models, ignoring any level changes due to the Zika Virus epidemic. Black dashed lines show the estimated deseasonalised expected (counterfactual) number of live births had the COVID-19 pandemic never happened and pre-pandemic trends continued instead. Socioeconomic indicators used are shown in parentheses on top of each panel. For Spain, only births with maternal age over 25 are included (see methods and suppl. material).

Sweden, there was (almost) no decrease in the number of live births at the start of the exposed period (Figs. 2, 3). Except for Brazil, there was a subsequent increase in live births that resulted in a higher average weekly/monthly number of live births in these countries during the exposed period. There were fewer babies born to socioeconomically advantaged groups in Scotland, the United States, and especially Spain at the beginning of the exposed period (blue dots near the vertical line in Fig. 2). However, the subsequent rebound in the number of births outweighed the initial drop.

In Fig. 4 and Supplementary Table 1, we show the differences between the socioeconomic composition of the observed cohort of live births conceived during the pandemic and their respective coun-terfactual compositions for each country. In Austria, England, Finland, the Netherlands, Scotland, Spain, and the United States, the propor-tion of babies born to socioeconomically advantaged groups during December 2020 – December 2021 was higher compared with their counterfactual cohort composition.

In Brazil, Colombia, Ecuador, and Mexico, the proportion of live births born to socioeconomically advantaged groups decreased (Fig. 4). These were also the only included countries where the pro-portion of live births born to socioeconomically disadvantaged groups increased (Fig. 4), except for Denmark. Denmark had a 0.5% [95%CI: 0.1%; 0.9%] increase in the share of babies born to women in the bot-tom 20% of equivalised household income. This result was not con-firmed when using maternal educational attainment or non-

equivalised household income (see Denmark's country profile in the suppl. material). In Wales, there was no change in the proportion of babies born to the socioeconomically advantaged groups but a decrease in the proportion of babies born to women living in the most socioeconomically disadvantaged areas (Fig. 4). Compared with the socioeconomic composition of their counterfactual cohorts, Spain saw the largest percentage point increase in the share of live births born to women with post-secondary and tertiary educational attainment (2.5% [95%CI: 2.2%; 2.9%]). In contrast, the share of live births with unknown maternal educational attainment decreased by -2.6% [95%CI: -2.8%; -2.4%]) while the share of babies born to women with primary or lower secondary maternal educational attainment did not change (Fig. 4).

For Austria, the Netherlands, and especially England and Scotland, these differences in socioeconomic composition approximated a socioeconomic gradient: Starting with a decrease in the proportion of live births among the most socioeconomically disadvantaged, each step towards the least disadvantaged group was associated with a smaller decrease or higher increase. The opposite socioeconomic pattern was visible for Brazil and Ecuador.

It is worth noting that the proportion of live births with missing information on socioeconomic circumstances also changed by at least half a percentage point for Austria (−0.5% [95%CI: −0.8%; −0.2%]), Denmark (0.7% [95%CI: 0.6%; 0.8%]), Colombia (−0.8% [95%CI: −0.9 %; −0.7%]), and Spain (−2.6% [95%CI: −2.8%; −2.4%]), (Fig. 4, Supplemen-tary Table 1).

## Difference between Observed and Counterfactual Socioeconomic Composition of Live Births December 2020-December 2021

Absolute Difference in the Proportion of Live Births between Observed vs. Counterfactual Composition

**Fig. 4 | Percentage point differences between the observed and counterfactual socioeconomic composition of the December 2020–2021 birth cohort.** The black dots are percentage-point differences derived from the estimated mean counterfactual number of live births using interrupted time series Poisson regressions on the number of live births. 95% confidence intervals presented as horizontal lines around the point estimates are estimated using a three-step approach (see method section). Estimates are based on 596,276 live births for Austria; 19,995,228 for Brazil; 4,329,577 for Colombia; 413,903 for Denmark; 1,965,574 for Ecuador; 4,391,999 for England; 330,063 for Finland; 12,876,945 for Mexico; 1,167,024 for the Netherlands; 343,379 for Scotland; 134,874 for South Australia; 2,229,935 for Spain;

758,034 for Sweden; 22,306,054 for the United States; 216,797 for Wales. For Spain, only births with maternal age over 25 are included (see methods and suppl. material). Quintiles of equivalised household income and area deprivation are abbreviated by the letter Q. Q5 (highest income, lowest deprivation) denotes the fifth quintile, Q4 the fourth quintile, and so forth. For countries, where only parental educational attainment was available, "Post-Second./ Tert." denotes post-secondary or tertiary educational attainment, "Upper Second." denotes upper secondary educational attainment, "Lower Second." denotes lower secondary educational attainment, and "Prim." denotes primary educational attainment.

Complementary to differences between observed and counterfactual proportions in the main manuscript, we present the relative differences between the observed and counterfactual number of births by socioeconomic group and country in our supplementary material (Supplementary Fig. 2). There, we also present the results of our re-estimation using alternative socioeconomic indicators (Supplementary Figs. 3, 4) Additionally, we show these differences for the maternal age and parity composition in Supplementary Figs. 5–8 where these data were available to us. In Ecuador, Finland, the Netherlands, Scotland, Spain, Sweden, and the United States, the observed cohorts tend to have older mothers than their counterfactual cohorts (Supplementary Fig. 5). In Brazil, Colombia, and Mexico, we observed the opposite. In Austria, there were no changes in the maternal age composition. The proportion of firstborns was lower than their counterfactual proportions in all countries except for Brazil and Denmark (Supplementary Fig. 7, no data were available to us for England, Scotland, South Australia, Sweden, and Wales).

## Discussion
### Summary
In this comparative study, we used population-wide administrative data (2015-2021) from 15 countries to compare the observed country-specific socioeconomic compositions of live births between December 2020 and December 2021 with their counterfactual compositions estimated by assuming that pre-pandemic trends and seasonal patterns in the number of live births would have continued in the COVID-19 pandemic period. In total, our study covers data of over 77.9 million

live births, out of which over 10.9 million were conceived after the first lockdown measures were introduced in March 2020.

For seven out of nine included European countries (Austria, England, Finland, the Netherlands, Scotland, Spain, Wales) and the United States, we found that babies conceived during the pandemic had more socioeconomically advantaged parental composition than expected had pre-pandemic trends continued. For all Latin American countries (Brazil, Colombia, Ecuador, and Mexico) included in our analysis, we found the opposite: the composition of live births conceived during the pandemic changed towards more socioeconomically disadvantaged groups. The percentage point differences in the proportion of live births born to socioeconomically advantaged groups were between −1% and +1%, except for Spain (+2.5%) and Ecuador (−3%). Similarly, the percentage point differences in the proportion of babies born to socioeconomically disadvantaged groups were between -1% and +1%, except for Colombia (+2.6%). For Denmark, South Australia, and Sweden, we found little to no evidence for changes in the parental socioeconomic composition during the pandemic.

**Comparison with previous literature on socioeconomic differences in fertility during the COVID-19 pandemic**
Socioeconomic differences in the fertility response to the pandemic have recently been confirmed in country-specific analyses of Spain[21], Norway[22], Iceland[31], Sweden[23], the United States[24,25], Colombia[26], Brazil[26], and Australia[28]. In Brazil and Colombia, fertility was found to have decreased for women with at least 8 years of schooling while this

effect was null or positive for women with fewer years in education[26]. In Spain[21], Norway[22], and the United States[25], more babies than expected were born to women with tertiary educational attainment, and a decrease (Spain) or little to no change in live births among women with lower educational attainment (Norway, United States). Similarly, in Iceland, the fertility increase in 2021 was mainly driven by the increase of third births - especially among women with tertiary educational attainment and high income[31]. Australia also experienced an overall increase in birth rates, but areas with higher unemployment, lower incomes, and a larger share of non-English speaking residents showed a slower growth in birth rates[28]. For Sweden, Ohlsson-Wijk and Andersson[23] found an increase in the relative risk of first and second births among Swedish-born mothers with 'high income' compared to Swedish-born mothers with 'medium income' in 2020. However, a comparison between their and our results are difficult because Ohlsson-Wijk and Andersson[23] only provide age-adjusted results for Swedish-born women and first and second births by different labour market activity while we estimate compositional changes in the entire birth cohort by equivalised household income, maternal educational attainment, and maternal age.

We add to this demographic evidence by specifically estimating percentage-point changes in the socioeconomic composition of entire birth cohorts conceived and born during the COVID-19 pandemic in a comparable approach across 15 countries. This wider focus is warranted by the under-recognised potential effect of sudden compositional changes of births on population-level differences in outcomes that are associated with socioeconomic circumstances at birth and/or parental socioeconomic position. Moreover, our comparative design enables us to abstract generalisable mechanisms from similarities and differences in the results across countries.

Our analysis was motivated by the Lockdown Cohort (LoCo) – effect hypothesis[20], which suggests that COVID-19 pandemic-induced compositional changes in the sociodemographic composition of parents may produce differences in life course outcomes between the birth cohort conceived during the pandemic (the LoCo) and earlier and later birth cohorts – even if early life exposure to the COVID-19 pandemic had no direct effect on life course outcomes. Independent of how socioeconomic characteristics at (or before) birth were measured – by household income, area-level deprivation, or parental educational attainment – we indeed found that the birth cohort conceived during the pandemic has a different socioeconomic composition than expected in twelve of fifteen included countries. While this general prediction is in line with our results, the initial formulation of the LoCo-effect hypothesis only considered change towards a more socioeconomically advantaged birth cohort that potentially explains counterintuitive improvements in health outcomes at birth during the pandemic[20]. Here, we show that shifts to a more socioeconomically advantaged birth cohort than expected were only observed in European countries and the United States. Our results of opposite changes in Latin American countries make it clear that considering national social contexts is essential for understanding changes in the socioeconomic composition of babies born and conceived during the COVID-19 pandemic.

Changes in reproductive behaviours during the COVID-19 pandemic could be explained by using various theoretical perspectives. For example, seen through the lens of Planned Behaviour Theory[39], the pandemic may have changed the intention to have a child because of potential negative consequences (e.g., increased health risks for pregnant women and offspring), a perceived increase in 'interfering' factors (e.g., economic uncertainty, income losses, or restricted access to reproductive technology), or perceived increases in 'enabling' factors (e.g., work-life balance, or lower opportunity costs). To try to explain the differences in compositional change between the included Latin American countries, the European countries, and the United States, it is helpful to draw from quantitative and qualitative research

on socioeconomic differences in fertility during the 2015-2016 ZIKV epidemic. In January 2016, several Latin American countries made official public recommendations to postpone pregnancy for 6 months to 2 years to avoid the risk of the Congenital Zika Syndrome[40]. Subsequent analyses of fertility using Brazilian data showed that, 9 months after the public recommendations to postpone pregnancies, declines in age-specific fertility rates were larger for women with higher educational attainment compared with any other groups[41]. As an explanation for this pattern in Brazil, there are stark socioeconomic inequalities in women's ability to postpone pregnancy – even in the face of the ZIKV epidemic or the COVID-19 pandemic[40,42,43]. These inequalities are produced by sociocultural norms surrounding gender roles and motherhood, access to contraceptives, and safe abortion procedures. As a consequence, women in more disadvantages socioeconomic positions have less fertility decision-making agency and a higher rate of unplanned pregnancies[40,42,43]. The 2015-2016 ZIKV epidemic might not only be useful as a comparison to the COVID-19 pandemic, but, in fact, may have contributed to the compositional changes we observed. In 2021, Marteleto and Dondero[43] reported that women in Brazil still have a lively memory of the health risks associated with pregnancy during the ZIKV epidemic. In their sample, more than 75% of women cited "enormous amounts of fear and worry about COVID-19 and its health and economic consequences as reasons for wanting to delay or avoid pregnancy" in in-depth interviews. Furthermore, as a preliminary result of their ongoing work, the authors report that 90.2% of their survey participants think that women should not get pregnant during the COVID-19 pandemic[43]. Therefore, it is plausible that the ZIKV epidemic may have amplified any effects of the COVID-19 pandemic on the socioeconomic composition of birth cohorts in regions previously affected by the ZIKV epidemic.

There are also plausible explanations for why we found no changes in socioeconomic composition in South Australia and Sweden. If we assumed that increased economic uncertainty and strict lockdown measures caused changes in the socioeconomic composition of birth cohorts, then we would expect to see no compositional changes in the absence of these factors. Our finding of no compositional changes in South Australia, where strict lockdown measures, such as 'stay-at-home orders' only lasted for a few weeks and generous income support schemes counteracted the economic collateral damage of the pandemic, supports this explanation. Similarly, the results for Sweden, which pursued a pandemic mitigation policy without strict lockdown measures (and instead relied on individuals to take measures to avoid transmission), paired with substantial reinforcements of the social protection system to protect incomes and living standards[44], suggest that policy responses can influence the way socioeconomic birth cohort composition reacts to macro-level shocks. Unlike Sweden, Denmark did pursue a more extensive lockdown strategy but offered substantial economic mitigation packages[45], and only saw a marginally larger decline in consumer spending than neighbouring Sweden[46], indicating the potential importance of financial security for fertility decisions during the pandemic.

## Generalising the findings

To summarise, the direction of change in the socioeconomic composition induced by the COVID-19 pandemic, and potentially other macro-level shocks, crucially depends on how a socioeconomic position relates to agency in fertility decision-making. That is, in contexts where women in socioeconomically disadvantaged positions have less control over their fertility decisions than women in socioeconomically more advantaged positions, macro-level shocks that exert a downward pressure on fertility across the population will cause compositional change towards a more socioeconomically disadvantaged birth cohort. The ZIKV epidemic in Brazil is a useful example of a population-wide downward pressure on fertility to which this mechanism applies. The COVID-19 outbreak in Latin American countries may have also

produced population-wide downward pressure on fertility due to the heightened risk perception caused by the preceding ZIKV epidemic[43]. In the context of public health emergencies, this mechanism linking population-wide downward pressure on fertility and change in socioeconomic composition of birth cohorts holds even if there are no socioeconomic differences in risk aversion during pregnancy and even if there are no socioeconomic inequalities in the risk of harm.

In contexts of higher agency in fertility decisions and lower inequalities therein, fertility can be more responsive to public health emergencies or other macro-level shocks. Here, the socioeconomically unequal distribution of shock-induced adversities and/or benefits will lead to a more socioeconomically advantaged birth cohort as suggested by the LoCo-effect hypothesis. This assumes that shock-induced downward pressures on fertility outweigh upward pressures among socioeconomically disadvantaged groups and/or shock-induced upward pressures on fertility prevail among socioeconomically advantaged groups[20]. For a list of previously hypothesised potential upward and downward pressures of the COVID-19 pandemic on fertility and according to theories, please see relevant papers focused on fertility[19,36,39,47,48].

### Which groups are driving the compositional change during the COVID-19 pandemic?

For Brazil, Colombia, Ecuador, England, Scotland, Spain, the United States, and Wales, we found changes in the number of births among either the socioeconomically disadvantaged or advantaged groups and no changes among other groups. For Austria, Finland, Mexico, and the Netherlands, on the other hand, births in all socioeconomic groups either increased or decreased, but compositional changes were produced by differences in the size of these changes. Among countries where we found a socioeconomically more advantaged birth cohort than expected (except for Spain), this compositional change was driven by increases in the number of live births among advantaged groups rather than decreases in the number of live births among socioeconomically disadvantaged population groups. Put differently, compositional change in these countries was driven by socioeconomically advantaged groups having more babies than expected, rather than socioeconomically disadvantaged groups having fewer babies than expected.

### Are these compositional changes impactful?

To understand the potential impact of our results on life course outcomes, it may be helpful to interpret the size of these compositional changes in two ways.

First, we draw a comparison with prominent research that used a cross-cohort comparison strategy to show that in-utero exposure to the 1918 Influenza epidemic in the United States was associated with lower adult socioeconomic position and increased rates of physical disability in adult life compared with birth cohorts conceived before and after the 1918 Influenza epidemic[6,49]. In 2022, Beach et al.[38] published a study that used United States enlistment records during the First World War and census data including information on the parental characteristics of the birth cohort exposed to the 1918 Influenza epidemic in utero to investigate if often-cited effects[6,49] persist after controlling for potential compositional changes in parental characteristics. Beach et al. showed that fathers of the exposed 1919 birth cohort had, on average, a lower socioeconomic position compared with fathers of earlier and later cohorts. This difference in paternal socioeconomic composition between the 1918-1920 birth cohorts existed because men in higher socioeconomic positions were more likely to be drafted for military service during the first world war[38]. For example, fathers of the exposed 1919 birth cohort were composed of 91% literates while fathers of the 1918 birth cohort were composed of about 91.6% literates (see Fig. 2 in Beach et al.)[38].

The compositional changes we found for the proportion of babies born to the most socioeconomically advantaged and disadvantaged groups in the December 2020 - December 2021 birth cohort are comparable to and often exceed this 0.6% percentage point difference in literate fathers induced by the 1918 Influenza epidemic and the First World War in the United States. For example, we estimated that, in the United States, the birth cohort conceived during the COVID-19 pandemic is composed of 1% [95%CI: 0.9%; 1.1%] more babies born to women with post-secondary or tertiary educational attainment than their counterfactual cohort would have had if pre-pandemic trends had continued. Although the effect of paternal illiteracy on life course outcomes in the 1919 birth cohort might not be comparable to the effect of parental tertiary educational attainment in the December 2020 – December 2021 birth cohort, this puts our effect sizes for compositional change into a historical perspective.

Importantly, the size of these compositional differences in the case of the 1918 Influenza epidemic was strong enough to substantially attenuate previously found effects of in-utero exposure on adult socioeconomic outcomes once parental characteristics were controlled for, rendering the effect of in-utero exposure statistically insignificant[38]. In other words, the differences in adult life socioeconomic position between the 1919 birth cohort and its adjacent birth cohorts were more likely attributable to differences in parental characteristics than to direct in-utero exposures. The size of compositional changes we found may therefore be large enough to significantly bias similar studies of early life exposure to the COVID-19 pandemic on life course outcomes.

Second, the extent to which compositional changes in parental socioeconomic position may produce differences in life course outcomes between the cohort conceived during the pandemic and surrounding birth cohorts is contingent on the strength of association between the socioeconomic positions of parents and the outcomes of interest. For example, although there are socioeconomic inequalities in birth outcomes like preterm birth or low birth weight[34,50], the size of currently observed inequalities in these outcomes together with observed compositional changes are unlikely to produce substantial differences in birth cohort averages. For life course outcomes with larger inequalities by parental socioeconomic position, like completion of tertiary educational attainment, the observed compositional changes might indeed produce detectable differences between birth cohorts like those found for the 1919 birth cohort conceived during the 1918 Influenza epidemic[38,49].

Importantly, the effects of compositional changes and potential direct effects of in-utero or early life exposure to the pandemic on life course outcomes are not mutually exclusive. Still, it is plausible that parental composition did not only change in observable characteristics like, e.g., age at birth, parity, socioeconomic position, residential area, or ethnicity, but also in unobserved parental characteristics that might be associated with future outcomes of interest. Future studies using the COVID-19 pandemic as in-utero or early life exposure will have to carefully consider their analytical strategies to avoid bias associated with compositional change in observed and unobserved parental characteristics.

### Strengths

The main strength of our study is its comparative design because it allows us to target a specific statistical estimand and use the same methodological approach to estimate it across 15 countries by using birth register data with almost complete coverage of births. In addition to providing comparable evidence for pandemic-induced compositional change across countries, this study design enables us to present a powerful empirical foundation for generalisable population-level mechanisms based on similarities and differences in results across countries. Although the effect size of compositional changes varied across countries, the consistency in similarities between European

countries and the United States and in the differences between these countries and the Latin American countries supports our proposed mechanisms behind macro-level shocks and corresponding changes in birth cohort composition. Further, changes in socioeconomic composition were visible irrespective of whether we used maternal or parental educational attainment, household income, or area-level deprivation of the mother's residential area as an indicator of socioeconomic circumstances. The deviation of results for the exceptional pandemic cases, Sweden and South Australia, gives additional credibility to our results and their interpretation.

## Limitations

In our counterfactual scenario, we assumed that pre-pandemic trends in the population-group-specific number of live births had continued in the absence of the COVID-19 pandemic. Alternative counterfactual scenarios could have assumed abrupt changes in the level of the number of births, a sudden reversal in secular trends (for example, an increase after a long decline) in the number of births, and/or changes in the seasonality of birth counts for some or all population groups during the 13-month period between December 2020 and December 2021. However, in our view, these counterfactuals would require stronger justification than our approach and weaken comparability across countries.

In Austria, Denmark, Colombia, Mexico, the Netherlands, and Spain, the differences between the observed and counterfactual number of live births without information on socioeconomic circumstances were non-negligible when compared to differences in the number of births in the other population groups. Thus, increases or decreases in the number of births for population groups could be affected by missing data or misclassification of parental socioeconomic position. Another alternative explanation for observed compositional changes, among included countries (especially Ecuador and Mexico), where delayed registration of birth is common, could be that the registration of births was drastically delayed for some population groups during the COVID-19 pandemic. However, if delayed registration of births was more common among socioeconomically disadvantaged population groups in these countries, the inclusion of these births would lead to an even stronger estimated compositional shift towards a socioeconomically more disadvantaged birth cohort than our estimates show. Where relevant, the issue of delayed birth registration is discussed in more detail in the country profiles (see suppl. material).

Although in the Netherlands, Spain, Denmark, Sweden, Colombia, and Mexico the proportions of missing socioeconomic information ranged from 2.1% to 5.7%, these alternative explanations are unlikely, because in other included countries (Brazil, Ecuador, England, Scotland, and Wales) with very little missing information on area-level deprivation of mothers' residential areas, changes in socioeconomic composition showed an overall consistent pattern. Moreover, included countries with near-complete population coverage for 2021 showed the same pattern in compositional changes as observed for countries with lower coverage and delayed birth registrations, e.g., Colombia, Ecuador, or Mexico. Lastly, the number of births with missing socioeconomic information was substantially reduced for Austria and Spain, once we used the highest available non-missing parental educational attainment instead of maternal educational attainment (see suppl. material). Using this alternative indicator of parental socioeconomic circumstances, percentage point differences between observed and counterfactual proportions of births with missing information shrank towards zero, and the results reinforced compositional change towards a more socioeconomically advantaged birth cohort in Austria and Spain.

Naturally, we would have liked to have better comparable measures of socioeconomic circumstances of live births to make more valid cross-country comparisons of effect sizes. Having readily

available and validated area-level material deprivation indices that can be linked to geographical information available in birth registers would be a simple way of increasing homogeneity of socioeconomic indicators in cross-country studies – especially in countries where socioeconomic information contained in population registers is of limited quality or cannot easily be linked to birth registers or other health-related data.

We do not take an intersectionality approach to trace how pre-existing social inequalities and evident intersectional inequalities in the COVID-19 pandemic's impact lead to compositional changes in the December 2020 - December 2021 birth cohort[16]. Such estimates will require careful country-specific studies with registers that contain reliable information on ethnicity.

We observed compositional change towards higher maternal age for Ecuador, Finland, the Netherlands, Scotland, Spain, Sweden, and the United States. As higher socioeconomic position is associated with higher maternal age, this result is not surprising. Still, the differences between the socioeconomic composition of the observed and counterfactual December 2020 - December 2021 birth cohort persisted even among women aged 26 and older in Spain or in Austria, where we observed compositional changes regarding socioeconomic circumstances but not regarding maternal age. Moreover, the December 2020 - December 2021 birth cohort is composed of fewer firstborns than expected in Austria, Colombia, Ecuador, Finland, Mexico, the Netherlands, Spain, and the United States. Future research that investigates the extent to which this compositional change may be an artefact of compositional change towards higher maternal age or a result of pandemic-induced heightened uncertainty would be useful to better understand these compositional changes in parity.

Lastly, we lack the data to tell if compositional changes reverse in subsequent birth cohorts or if the COVID-19 pandemic led to a lasting change in birth cohort composition. Evidence from Spain suggests that fertility recuperation differs by age group and parity[51]. The finding that the birth deficit among women under 25 was still increasing in Spain by the end of 2021 might indicate that socioeconomically disadvantaged groups take longer to change their fertility intentions, or that (perceived) economic uncertainty has not yet improved for these groups[51].

## Implications of this study

In twelve out of fifteen included countries – Austria, Brazil, Colombia, Ecuador, England, Finland, Mexico, the Netherlands, Scotland, Spain, the United States, and Wales – we found evidence for changes in the parental socioeconomic composition of the birth cohort (December 2020 – December 2021) conceived during the pandemic. Where material barriers and sociocultural norms make it more difficult for women in socioeconomically disadvantaged positions to postpone pregnancy in the face of adversity, we observed pandemic-induced compositional changes towards a more socioeconomically disadvantaged birth cohort. Conversely, in contexts with higher agency in fertility decision-making, we observed compositional changes toward a less socioeconomically disadvantaged cohort. In South Australia and Sweden, where lockdown measures were relatively short-term to non-existent and governments provided generous economic support to mitigate increased economic uncertainty, we found no evidence for changes in the socioeconomic composition. Denmark, which had semi-stringent lockdown measures and generous economic support, also saw little evidence for change.

Socioeconomic position of parents and socioeconomic circumstances at birth are strong predictors of many important health, developmental, and socioeconomic outcomes throughout the life course[33,34]. Thus, observed compositional changes during the pandemic may well produce between-cohort differences in life course outcomes with strong socioeconomic gradients such as educational attainment, income, health, or mortality. Researchers must keep these compositional changes in mind when interpreting such between-

cohort differences and when aiming to estimate the effect of in-utero or early life exposure to the pandemic on life course outcomes. As cohorts conceived during the COVID-19 pandemic and surrounding birth cohorts age, their between-cohort differences in parental socio-economic composition and life course outcomes will offer valuable opportunities to advance knowledge on how public health emergencies and other macro-level shocks affect human populations.

## Methods

### Data

We used population-wide birth register data from 15 countries spanning the period 2015 to 2021. For Austria, Brazil, Colombia, Denmark, Ecuador, Finland, Mexico, the Netherlands, South Australia, Spain, and Sweden, we had access to individual-level data. We used these data to create time series of the weekly or monthly number of live births by available indicators of parental socioeconomic circumstances (equivalised household income, maternal level of educational attainment, area-level deprivation). For England, Scotland, the United States, and Wales, we obtained aggregated monthly time series data on live births by available socioeconomic indicators. Where feasible, we created weekly time series of live births (Brazil, Ecuador, Denmark, Finland, Mexico, Sweden). Otherwise, we used monthly time series data on the number of live births.

For Austria, restricted access to anonymous individual-level birth data was purchased by TW from Statistics Austria and used to create aggregated monthly time series. Ethical approval to use administrative data for research purposes is covered by the Austrian (Bundesstatistikgesetz 2000).

For Brazil, we used openly available anonymous individual-level data from SINASC (Sistema de Informações sobre Nascidos Vivos) (https://opendatasus.saude.gov.br/dataset/sistema-de-informacao-sobre-nascidos-vivos-sinasc) and the openly available small-area deprivation index BrazDep (doi: 10.36399/gla.pubs.215898). Ethical approval to use the publicly available data for research purposes is covered by national legislation (Resolution No. 510, 2016 of the National Health Council of Brazil).

For Colombia, we used openly available anonymous individual-level data used from DANE (Departamento Administrativo Nacional De Estadística) (https://microdatos.dane.gov.co/index.php/catalog/DEM-Microdatos#_r=1700214975152&collection=&country=&dtype=&from=2015&page=1&ps=&sid=&sk=&sort_by=title&sort_order=&to=2023&topic=&view=s&vk=). Ethical approval to use anonymous individual-level publicly available data from DANE for research purposes is covered by national legislation (Law No. 2335 of 2023).

For Denmark, we used restricted access pseudonymised individual-level birth data provided by Statistics Denmark under auspices of project 703566. Ethical approval to access the data for research purposes are covered by the data access permit granted to project 703566.

For Ecuador, we used openly available anonymous individual-level vital statistics from INEC (Instituto Nacional de Estadística y Censos) (https://aplicaciones3.ecuadorencifras.gob.ec/BIINEC-war/index.xhtml) and the area-level deprivation index (doi: 10.11606/s1518-8787.2019053001410) received upon request from Andrés Peralta. INEC is the national statistics institution of Ecuador, and its public distribution of birth data is in compliance with national legislation regulating the use of these data for research purposes.

For England, we purchased monthly time series data (2015-2022) of the number of live births by deciles of the Index for Multiple Deprivation (IMD) from the Office for National Statistics. These data are now openly available. (https://www.ons.gov.uk/peoplepopulationandcommunity/birthsdeathsandmarriages/livebirths/adhocs/1703livebirthsbymonthofoccurrenceandimddecileenglandandwales2015to2022).

For Finland, we used restricted-access pseudonymised individual-level birth data provided by Statistics Finland under access permits TK/

1170/07.03.00/2023 and THL/6303/14.06.00/2023. Ethical approval to use these data for research purposes is covered by the data access permits.

For Mexico, we used openly available anonymous individual-level data INEGI (Instituto Nacional de Estadística y Geografía) (https://en.www.inegi.org.mx/programas/natalidad/#microdata). INEGI is the national statistics institution of Mexico and their public distribution of individual-level birth data is in compliance with the national regulatory framework regulating the use of these data for research purposes as described on their website (see https://en.www.inegi.org.mx/programas/natalidad/#microdata).

For the Netherlands, we used restricted access pseudonymised individual-level birth data provided by Statistics Netherlands under access permit for project number 8552. Ethical approval to use administrative data for research purposes is covered by Dutch law (Wet medisch-wetenschappelijk onderzoek met mensen).

For Scotland, we used openly available time series data from the Scottish Morbidity Record 02 (https://scotland.shinyapps.io/phs-covid-wider-impact/).

For South Australia, we used restricted access pseudonymised individual-level data from the Better Evidence Better Outcomes Linked Data (BEBOLD) platform via approval by the SA Department for Health and Wellbeing Human Research Ethics Committee (2022/HRE00137) and The University of Adelaide Human Research Ethics Committee (37934).

For Spain, the openly available anonymous individual-level birth data provided by the Instituto Nacional de Estadística (INE) (https://www.ine.es/dyngs/INEbase/en/operacion.htm?c=Estadistica_C&cid=1254736177007&menu=resultados&secc=1254736195443&idp=1254735573002#!tabs-1254736195443). Dissemination of these publicly available anonymised data by INE is in compliance with national legislation (Law 12/1989 on the Government Statistics Act).

For Sweden, we used restricted-access pseudonymised individual-level data through participation in the SWECOV project. The ethical permit was granted by the Swedish Ethical Review Authority (Permit 2024-02342-02).

For the United States, we used openly available aggregated time series data from Centers for Disease Control and Prevention WONDER portal (https://wonder.cdc.gov/natality.html).

### Indicators of parental socioeconomic circumstances

As primary indicators of parental socioeconomic circumstances, we used equivalised household income, pre-existing measures of (small) area-level deprivation, and maternal educational attainment as available in population registers or on birth certificates. For equivalised household income (Denmark, Finland, the Netherlands, Sweden) and area-level deprivation (Brazil, Ecuador, England, Scotland, South Australia, Wales), we used pre-pandemic measures, so that pandemic-induced effects on the parental socioeconomic composition cannot be driven by pandemic-induced changes in the income distribution or deprivation measure of an area. Live births were categorised into quintiles of equivalised household income and area-level deprivation. Maternal educational attainment at birth or in the year of birth was used for Austria, Colombia, Mexico, Spain, and the United States. For comparative visualisations, we grouped levels of educational attainment into primary and lower secondary (or compulsory), upper secondary, and post-secondary and tertiary educational attainment of the mother (except for Mexico, due to availability, see suppl. material).

The availability and quality of data on indicators for parental socioeconomic circumstances varied across countries, and it is important to note that the socioeconomic indicators are not directly comparable. For example, although we used area-level measures of deprivation for Brazil, Ecuador, England, Scotland, South Australia, and Wales, their measurement varied. These area-level measures

combine different indicators, are measured on different spatial levels, and use data from different years.

Missing data on socioeconomic indicators at birth varied across countries. Over the entire period, there were less than 1.5% live births without information on parental socioeconomic circumstances in Austria, Brazil, Ecuador, England, Finland, the United States, Scotland, South Australia, Wales, and 1.8%, 2.6%, 3.8%, 3.9%, 5.8%, 6.6%, and 14.3% missing information on maternal educational attainment in Finland, the Netherlands, Denmark, Colombia, Sweden, Mexico, and Spain respectively. As educational attainment levels in Spanish Vital Statistics are only assigned to people over 25, this was our target population for Spain (see suppl. material p.78 and onwards). In this group, 4.4% of live births between 2016 and 2021 had missing information on maternal educational attainment. For the United States and Spain, data on the number of live births by educational attainment only included data from 2016-2021 because not all United States reporting regions provided information on maternal education in 2015, and because Spain changed their collection of educational attainment in 2015, so that levels are hardly comparable across time (see respective country profiles in suppl. material).

In sensitivity analyses, we used alternative or more detailed indicators of parental socioeconomic circumstances where available. For countries with maternal educational attainment as the primary indicator, we also used the highest educational attainment of both parents (Austria, Colombia, Mexico, Spain) or paternal educational attainment (United States) in complementary analysis.

In our supplementary material, we describe the data sources, measurements, and missingness in more detail for each country separately (see country profiles). We additionally present country-specific results on the compositional change in absolute and relative terms regarding alternative socioeconomic indicators, maternal age, and parity where available.

## Birth cohort exposed to potential compositional change

For our included countries, strict lockdown measures to contain the spread of COVID-19 were introduced in mid-March 2020 as indicated by the Stringency Index (Oxford COVID-19 Government Response tracker[52], see Supplementary Fig. 1). Although some babies conceived in the second half of March 2020 have been born before December 2020, we assume, based on the average gestational age at birth, that live births conceived during the pandemic will present a substantial share (~50%) of live births in the second half of December 2020. Therefore, we set the start of the exposed period to December 2020 for monthly time series and the start of the 51st week of 2020 for weekly time series (Supplementary Fig. 1).

Using birth cohorts over conception cohorts entails misclassification of births close to the start of our exposure period. There are babies born just after our cutoff (51st week of 2020 or December 2020) that have been conceived before the onset of lockdown measures in mid-March 2020. Because of anticipation, this group of births may have still been exposed to pandemic-induced changes in fertility. They mostly consist of full-term births because these pregnancies had to last at least from before the onset of lockdown measures (~11th week of 2020) to our chosen cutoff for the exposure period (51st week of 2020). This may select for maternal characteristics that are protective for preterm birth like maternal age[53] or more advantaged parental socioeconomic circumstances[34,50]. Conversely, there is an upper bound of gestational age (~39 weeks) for babies conceived just after the onset of lockdown measures and born just after our cutoff. This upper bound may select for parental characteristics that are associated with preterm birth. Further, there may have been direct effects of exposure to the pandemic on preterm births[54,55] possibly moderated by parental socioeconomic circumstances[54,56] and an indirect effect on preterm birth rates close to our cutoff through pandemic-induced drops in conceptions[12,21] which may have also been moderated by

parental socioeconomic circumstances[12,20]. These could be other reasons for why our chosen cutoff is not accurately distinguishing between live births conceived before and after the onset of lockdown measures.

However, the period vulnerable to misclassification is short and the number of potentially misclassified births is low. Thus, we believe that the advantages of using birth cohorts (knowledge on exact birth date, no missing data on gestational age, and no between-country variation in measurement) outweigh the disadvantages described above.

## Analytical strategy

Our main aim was to estimate compositional differences in parental socioeconomic circumstances between the cohorts conceived during the COVID-19 pandemic and their counterfactual compositions had pre-pandemic trends continued, and the pandemic never occurred.

To obtain these, we first estimated the level changes in the number of live births between December 2020 and December 2021 by interrupted time series Poisson regression models. These models were estimated for each socioeconomic group (e.g., the highest fifth of equivalised household income or post-secondary and tertiary maternal educational attainment) and country separately. We specified our models with linear and quadratic terms for time-trends and week/month of the year indicator variables to address for seasonality (January or the first week of the year as reference). For Brazil, Colombia, Mexico, and Ecuador we adjusted for the effect of the ZIKV epidemic on the number of births by including a binary variable indicating the period from August 2016 to December 2016 as informed by previous studies (for details, see country profiles in suppl. material)[41,57,58].

We used the parameters estimated by the models described above to estimate the sum of the group-specific number of births over the exposed period (December 2020 – December 2021) had the COVID-19 pandemic not happened and, counter to the fact, pre-pandemic trends continued instead[59]. This estimation of the counterfactual is justified as we only estimate the number of births for a short period of 13 months for monthly data or 54 weeks for weekly data. Assuming abrupt changes in the level, secular trend, and seasonality of the number of births between December 2020 and December 2021 in the absence of the pandemic would be less realistic.

Next, to estimate the counterfactual compositions, we calculated each group-specific proportion of live births by dividing the obtained counterfactual group-specific number of births by the sum of all group-specific counterfactual numbers in each country using our point estimates. To calculate the percentage point differences in proportions of births, we subtracted the counterfactual proportions from their respective observed proportions.

To obtain 95% confidence intervals for these differences in composition, we used a three-step approach. First, we drew a random counterfactual number of live births from a normal distribution with the mean equal to our point estimate for the group-specific counterfactual number of live births and the standard deviation equal to our estimate's standard error. The random draws for each group are independent, as we assumed that the absolute number of live births in one population group is independent of the number of live births in the other groups.

Second, we created a counterfactual cohort composition by dividing the randomly drawn group-specific numbers of live births by the sum of all group-specific random draws.

Third, we calculated the percentage point difference between the observed group-specific proportion of live births and the respective, randomly drawn, counterfactual group-specific proportion.

We repeated these three steps 10,000 times per country to obtain 10,000 different counterfactual cohort compositions and their respective differences with the observed composition. The lower and upper bounds of our 95% confidence intervals for the group-specific

differences between observed and counterfactual proportions of live births are then given by the 2.5th and 97.5th percentiles of each group-specific distribution.

Although selecting different modelling approaches for each country and parental characteristic could lead to slightly more accurate estimates when time series are non-linear and autoregressive beyond the accounted monthly and weekly seasonality, we prefer a simpler, homogenous approach for the sake of comparison and interpretability.

The number of women of reproductive age in each level of our parental characteristics is unavailable in our data sources for most countries. Therefore, we were not able to include an exposure variable in these Poisson regression models. As we are not trying to estimate changes in fertility rates but the socioeconomic composition of live births, this limitation is tolerable.

### Reporting summary
Further information on research design is available in the Nature Portfolio Reporting Summary linked to this article.

## Data availability
Data cannot be shared by us. We used openly accessible data for: Brazil from SINASC (Sistema de Informações sobre Nascidos Vivos) (https://opendatasus.saude.gov.br/dataset/sistema-de-informacao-sobre-nascidos-vivos-sinasc), Colombia from DANE (Departamento Administrativo Nacional De Estatdistica) (https://microdatos.dane.gov.co/index.php/catalog/DEM-Microdatos#_r = 1700214975152&collection = &country = &dtype = &from=2015&page=1&ps = &sid = &sk = &sort_by=title&sort_order = &to=2023&topic = &view=s&vk =), Ecuador from INEC (Instituto Nacional de Estadística y Censos) (https://aplicaciones3.ecuadorencifras.gob.ec/BIINEC-war/index.xhtml), Mexico from INEGI (Instituto Nacional de Estadística y Geografía) (https://en.www.inegi.org.mx/programas/natalidad/#microdata), Scotland from the Scottish Morbidity Record 02 (https://scotland.shinyapps.io/phs-covid-wider-impact/), Spain from INE (Instituto Nacional de Estadística) (https://www.ine.es/dyngs/INEbase/en/operacion.htm?c=Estadistica_C&cid=1254736177007&menu=resultados&secc=1254736195443&idp=1254735573002#!tabs-1254736195443), and the United States from the Centers for Disease Control and Prevention WONDER portal (https://wonder.cdc.gov/natality.html). We purchased the necessary data for England and Wales from the Office for National Statistics (ONS) which is now openly available on their website (https://www.ons.gov.uk/peoplepopulationandcommunity/birthsdeathsandmarriages/livebirths/adhocs/1703livebirthsbymonthofoccurrenceandimddecileenglanda§ndwales2015to2022). Links to the openly available data are also provided in the respective country profiles in our supplementary material. Links have been last accessed on the 18.09.2025. For Austria, Denmark, Finland, the Netherlands, South Australia, and Sweden, we used restricted access individual-level data. We are not allowed to share these data, but researchers can apply for data access through the respective data holders: Austria: Austrian Micro Data Center, Statistics Austria (https://www.statistik.at/en/services/tools/services/center-for-science/austrian-micro-data-center-amdc). Finland: Statistics Finland (https://guides.stat.fi/remote-access-to-research-data/using-microdata). Denmark: Statistics Denmark (https://www.dst.dk/en/TilSalg/data-til-forskning). Netherlands: Statistics Netherlands (CBS – Bureau voor de Statistiek), (https://www.cbs.nl/en-gb/our-services/customised-services-microdata/microdata-conducting-your-own-research/applying-for-access-to-microdata). South Australia: SANT DataLink (https://www.santdatalink.org.au/application_process). Sweden: Statistics Sweden (https://www.scb.se/en/services/ordering-data-and-statistics/microdata/) or through the SWECOV project (https://swecov.se/participation/).

## Code availability
Analyses were carried out in Stata v18[60] and all code for the statistical analysis is openly available at https://doi.org/10.5281/zenodo.15585751[61].

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

## Acknowledgements

We would like to thank Alastair Leyland, Anna Pearce, Andrea Tilstra and Ruth Dundas for their early contributions to the Lockdown Cohort-Effect hypothesis and for their involvement in initiating the research. We also thank Andrés Peralta for sharing the area-level deprivation index for Ecuador. MO's work was supported by the Marie Skłodowska-Curie Actions programme (grant agreement No 101150833), the Medical Research Council (MC_UU_00022/2), and the Scottish Government Chief Scientist Office (SPHSU17). HR, JL (Juha Luukkonen), MO, PTM were supported by the European Research Council under the European Union's Horizon 2020 research and innovation programme (grant agreement No 101019329) and grants to the Max Planck – University of Helsinki Center from the Jane and Aatos Erkko Foundation (Grant number 210046), the Max Planck Society (Grant number 5714240218), University of Helsinki (Grant number 77204227) and Cities of Helsinki, Vantaa and Espoo. The study does not necessarily reflect the Commission's views and in no way anticipates the Commission's future policy in this area. ESP is funded by the Wellcome Trust (225925/Z/22/Z). For related work, JVB got funding from ZonMw, NWO, Chiesi Pharmaceuticals, Strong Babies, de Snoo 't Hoogerhuys Foundation. MKR received funding from Stiftelsen Riksbankens Jubileumsfond (Ref. number: P23-0640). The funders had no role in the study design, data collection and analysis, decision to publish, or preparation of the manuscript. OMÖ received funding from the Swedish Research Council for Health, Working Life and Welfare (FORTE) (grant 2022-00262). AM, RP & JL (John Lynch) received funding from the National Health Medical Research Council of Australia (grants 1187489 & 1099422).

## Author contributions

M.O. conceived the comparative study, performed data preparation, data harmonisation and data analysis, produced the visualisations, drafted, and revised the manuscript and the supplementary material. H.R., J.L. (Juha Luukkonen) and P.T.M. provided critical feedback at all stages of the study and valuably edited all versions of the manuscript. P.T.M. was responsible for access to restricted Finnish data. J.M.L. contributed to the preparation of the Finnish data. T.W. was responsible for access to restricted Austrian data, performed the respective data preparation and provided feedback on multiple versions of the manuscript. L.B.-O., M.K.R. and J.V.B. developed a similar project for the Netherlands, which eventually merged with this study. L.B.-O., M.K.R. and J.V.B. were responsible for access to the restricted Dutch data and provided feedback on multiple versions of the manuscript. L.B.-O. performed the preparation of the Dutch data. O.MÖ. was responsible for access to restricted Swedish data, performed the respective data preparation and provided feedback on the revised version of the manuscript. P.F. was responsible for access to restricted Danish data performed the respective data preparation and provided feedback on the revised version of the manuscript. A.M., R.M.P. and J.L. (John Lynch) were responsible for access to restricted South Australian data, provided feedback on the revised version of the manuscript and A.M. performed the data preparation and data analysis for South Australian data. E.S.P., I.R.F., provided critical feedback on all versions of this manuscript.

## Competing interests

The authors declare no competing interests.

## Additional information

[1]Helsinki Institute for Demography and Population Health, Faculty of Social Sciences, University of Helsinki, Helsinki, Finland. [2]Max Planck - University of Helsinki Center for Social Inequalities in Population Health, University of Helsinki, Helsinki, Finland. [3]MRC/CSO Social and Public Health Sciences Unit, University of Glasgow, Glasgow, United Kingdom. [4]Department of Epidemiology, Center for Public Health, Medical University of Vienna, Vienna, Austria. [5]Tilburg School of Social and Behavioural Sciences, Tilburg University, Tilburg, Netherlands. [6]Department of Obstetrics and Gynaecology, Erasmus MC Sophia Children's Hospital, University Medical Centre Rotterdam, Rotterdam, Netherlands. [7]Department of Medical Epidemiology and Biostatistics, Karolinska Institutet, Stockholm, Sweden. [8]Division of Neonatology, Department of Neonatal and Paediatric Intensive Care, Erasmus MC Sophia Children's Hospital, University Medical Centre Rotterdam, Rotterdam, Netherlands. [9]Department of Public Health Sciences, Stockholm University, Stockholm, Sweden. [10]ROCKWOOL Foundation, Copenhagen, Denmark. [11]Swedish Institute for Social Research, Stockholm University, Stockholm, Sweden. [12]School of Public Health, The University of Adelaide, Adelaide, Australia. [13]Robinson Research Institute, The University of Adelaide, Adelaide, Australia. [14]Population Health Sciences, University of Bristol, Bristol, United Kingdom. [15]Center for Data and Knowledge Integration for Health (CIDACS), Instituto Gonçalo Moniz, Fiocruz Bahia, Fundação Oswaldo Cruz, Salvador, Brazil. [16]Faculty of Epidemiology and Population Health, London School of Hygiene and Tropical Medicine, London, United Kingdom. [17]Max Planck Institute for Demographic Research, Rostock, Germany. ✉e-mail: moritz.oberndorfer@helsinki.fi

