## [Transparent Peer Review file · Nature Communications]

The COVID-19 pandemic changed the parental socioeconomic composition of birth cohorts

Corresponding Author: Dr Moritz Oberndorfer

Version 0:

Reviewer comments:

Reviewer #1

(Remarks to the Author)

This paper provides a unified analysis of changes in the socioeconomic composition of mothers giving birth in 2021 (birth conceived during the pandemic) across 12 countries compared to countries' trends in prior years. The authors find that the socioeconomic composition of mothers increased in European countries and the United States while it decreased in Latin American countries.

The authors have put together a rich set of data spanning multiple countries and use a uniform framework to analyze compositional changes in mothers giving birth. The results are overall informative and convincing.

I have a few comments:

1. The main motivation of the paper that compositional changes in mothers giving birth have to be taken into account when studying outcomes among the cohorts conceived during the pandemic is well taken. However, this point can be made in one short paragraph rather than three pages.
2. Shortening this part will leave more space in the introduction to provide the paper's central motivation that is currently missing: given that there is plenty of evidence already of compositional changes among mothers conceiving during the pandemic from a large number of papers, what is this paper contributing? What are the remaining open questions in the field that the paper tries to answer?
3. In the discussion the authors write: "Socioeconomic differences in the fertility response to the pandemic have recently been confirmed in [several] country-specific analyses... We add to this literature by focussing on changes in the socioeconomic composition of cohorts conceived and born during the COVID-19 pandemic." But socioeconomic differences in the fertility response and changes in the socioeconomic composition of conceived cohorts are the same thing.
4. A more direct motivation for the paper could be to start with what is known about compositional changes; what are the remaining open questions; what this paper is doing; and only then discussing the implications for the study of pandemic cohort outcomes.
5. Figures 3 - 7 and Table 1 all show variations of the same information which is a bit confusing because as a reader one expects that different exhibits show different / new information. Also Figure 2 shows patterns that are very similar across all countries and this information can be summarized more efficiently in a sentence or two in the text (and the figure can be moved to the appendix). It is also not clear that Figure 1 is appropriate for the introduction since this paper is not about the effect of the pandemic on life course outcomes.

Reviewer #2

(Remarks to the Author)

NCOMMS-24-16743-T

The COVID-19 pandemic changed the socioeconomic composition of parents: A register-based study of 76.5 million live births in 12 countries

The authors of this article study how the COVID-19 pandemic influenced the socioeconomic composition of newborns. They draw from an exceptional amount of high-quality data from 12 high- and middle-income countries. They highlight the vast and heterogeneous impact of the pandemic on the cohort of newborns and draw implications for future studies, offering a cautionary tale for researchers examining this cohort of children.

The study is well-executed and provides a comprehensive and valuable effort to understand the consequences of the pandemic. The methodology is sound and well explained. The results align with other studies showing how patterns of selection into conception and child outcomes differed across contexts, such as in Spain and Chile. I believe the authors have done exceptional work, and I have no further comments. This study offers a substantial contribution to our understanding of the consequences of the COVID-19 pandemic.

Reviewer #3

(Remarks to the Author)

The work is of good quality in all its sections: the literature review, the quality of its data sources, the data analysis, and the discussion, which includes discussing the probable mechanisms behind its descriptive results. Above all, it successfully demonstrates what it aims to show: the change in the socioeconomic composition of births conceived during the COVID-19 pandemic.

Suggesting different mechanisms for different birth composition seems to me to be one of the strong points of the paper, derived from a solid theoretical assumption: that changes in a birth cohort's socioeconomic composition in response to macro-level shocks depend on how socioeconomic position relates to agency in fertility decisions

The solution found to the difficulties posed by the various indicators for parental socio-economic circumstances are also acceptable. However, it would be good to devote a little more space to commenting on the potential problems that the well-known delay in the timely annual registration of births during the pandemic, at least in some cases, such as Mexico, may cause. More space could also be devoted to justifying the choice of countries in the sample.

The solution to another common problem, generating a counterfactual scenario of expected births in the absence of a pandemic, is also well solved in the data analysis, although the paper will benefit from stating more clearly why assuming that "pre-pandemic trends continued instead" is the best option (in the context of fertility decline, when fertility reaches very low levels, a model could also indicate that the decline was going to stop or even 'rebound')

Recent Latin American and European scientific literature on the topic are well taken into account.

Finally, although the paper might not have space for great theoretical disquisitions, it would be good to mention that reproductive behaviour has been studied and modelled from various perspectives that could add theoretical thickness to studies on the subject, such as the Theory of Planned Behaviour or others.

Version 1:

Reviewer comments:

Reviewer #1

(Remarks to the Author)

Thank you for the detailed responses to my comments. I think the paper has become clearer and more concise.

I think it is still not clear what exact questions the paper tries to answer that have not been answered by the existing literature. The authors write that their they "bridge disconnected debates in demography on fertility responses to the pandemic" but what exactly these "disconnected debates" are is unclear. I agree with the editor and the other reviewers that there is scientific value in replicating and expanding existing analyses across a broad set of countries using a uniform methodological framework. But it is still important to clarify the exact priors from the existing literature.

Reviewer #2

(Remarks to the Author)

I congratulate the authors as the manuscript improved even more.

Reviewer #3

(Remarks to the Author)

I recommend the publication of the revised version

Version 2:

Reviewer comments:

Reviewer #1

(Remarks to the Author)

I have no more comments and recommend publication. Congratulations to a great paper.

We thank the reviewers for their time used to review our manuscript and their suggestions on how to improve the paper. We have responded to each comment below and marked changes to the manuscript in yellow. Further, we have used the formatting instructions to change the structure of the paper as required by the journal.

Please note that after submission in March 2024, our collaboration has grown, and we were able to add more data to the study after agreement of the journal's responsible editor. The revised manuscript now additionally includes results for Sweden, Denmark, and South Australia. Sweden and South Australia are especially interesting in this comparative study, as their pandemic experience and policy response to the pandemic was different to the other included countries. Data from Denmark, on the other hand, could add credibility to our existing theory. In the paper, we thus now additionally discuss what we may learn from the comparison with the exceptional cases Sweden and South Australia.

Reviewer #1

"This paper provides a unified analysis of changes in the socioeconomic composition of mothers giving birth in 2021 (birth conceived during the pandemic) across 12 countries compared to countries' trends in prior years. The authors find that the socioeconomic composition of mothers increased in European countries and the United States while it decreased in Latin American countries. The authors have put together a rich set of data spanning multiple countries and use a uniform framework to analyze compositional changes in mothers giving birth. The results are overall informative and convincing."

I have a few comments:

1. Comment

"The main motivation of the paper that compositional changes in mothers giving birth have to be taken into account when studying outcomes among the cohorts conceived during the pandemic is well taken. However, this point can be made in one short paragraph rather than three pages."

Response

Thank you for prompting us to be more concise. We shortened the introduction while trying to not lose clarity when presenting the analytical problem. We have **removed** Figure 1 and **removed** the following parts in the introduction shown in red.

Introduction

"The relatively clear period of population-wide exposure combined with available measures of exposure intensity (e.g. number of new COVID-19 cases or stringency of policy response²), offer researchers opportunities to estimate causal effects of living in a suddenly changed world regarding many outcomes."

"That is, the cohorts born and conceived before (or after) the pandemic need to be a useful representation of what would have happened to the cohort that was conceived and born during the pandemic had the pandemic not occurred. More specifically, the difference between the average

outcome of the exposed and the unexposed cohorts must not be confounded by another variable that causes the exposure and the outcome.”

“For example, the pandemic environment may have led to an increase in abortions among pre-pandemic conceptions in the most disadvantaged population groups due to deteriorating economic circumstances, while the rate of abortions did not change (or less so) in advantaged groups¹⁶ – assuming that there was equal access to safe abortions. Assuming increased barriers to safe abortions, the opposite scenario is also plausible.^{17–19} For babies born shortly after the start of the pandemic such selection mechanisms are of little concern²⁰, but these pregnancies might have been affected by disruptions in perinatal health services.^{12”}

“Our directed acyclic graph (DAG) in Figure 1 illustrates how these potential violations of the exchangeability assumption can bias estimation of the effects of in-utero and early life exposure to the pandemic even if there is no confounding variable that causes the exposure and the life course outcome of interest directly. The problem arises because analysing the effect of exposure to the pandemic inevitably conditions on observing the outcome. Trivially, a life course outcome, like birth weight of an individual can only be observed if that individual was conceived and survived all subsequent selection mechanisms (e.g., conception, miscarriage, abortion, stillbirth) until live birth. Thus, any effect of in-utero or early life exposures is partially mediated through potential pandemic-induced changes in fertility behaviour and/or survival^{14,15} until live birth. Because of this unavoidable conditioning on potential mediators, any variable (e.g., “Pre-pandemic Parental Characteristics” in Figure 1) that is associated with fertility behaviour and/or survival until live birth and the outcome will bias the estimate of the effect of in-utero or early life exposure to the pandemic even if this variable has no association with the exposure. This sort of bias is not entirely novel but contained in multiple well-known concepts: mediator-outcome confounding³¹, selection bias³², collider bias^{32,33}, immortal time bias, and survival bias^{14,15}

Figure 1: Directed Acyclic Graph (DAG) showing how conditioning on “Fertility Behaviour” and “Live Birth” (indicated by boxes) introduces mediator-outcome confounding of the effect of the “COVID-19 Pandemic” (exposure) on “Life Course Outcome” (outcome) of cohorts born and conceived during the pandemic. “Pre-pandemic Parental Characteristics” confound the mediator-outcome relationship between the mediators “Fertility Behaviour” and “Live Birth” and the outcome “Life Course Outcome”.

“These socioeconomic differences in the pandemic’s effect on fertility may have caused a change in the parental socioeconomic composition of live births conceived during the COVID-19 pandemic.”

2. Comment

“Shortening this part will leave more space in the introduction to provide the paper's central motivation that is currently missing: given that there is plenty of evidence already of compositional changes among mothers conceiving during the pandemic from a large number of papers, what is this paper contributing? What are the remaining open questions in the field that the paper tries to answer?”

Response

We thank the reviewer for these questions. We agree with the reviewer that the introduction has to make a stronger case about why our comparative approach is important despite the now existing evidence. With the help of the reviewer's comment, we now describe the central motivation of this paper as twofold. First, we want to give a (shortened) general description of the potential problems for future studies caused by compositional changes and analyse if such changes indeed happened. Second, by using a comparative design, to not only provide comparable evidence for compositional changes across multiple countries, but to also abstract generalisable macro-level mechanisms from similarities and differences in the results across countries.

This is now part of the revised introduction as part of the study aims (which have been rewritten according to the formatting guidelines) and in the revised discussion section where we discuss the strengths of the comparative study design in response to the reviewer's comment.

Introduction

“In this study, we use population-wide administrative data on 77.9 million live births from 15 countries covering the period 2015-2021 to analyse changes in the parental socioeconomic composition of babies conceived during the COVID-19 pandemic (born between December 2020 and December 2021). The contribution of this comparative study is twofold. First, we bridge disconnected debates in demography on fertility responses to the pandemic and in epidemiology on the pandemic's effect on pregnancy and birth outcomes by providing comparable evidence on compositional changes that can cause between-cohort differences in life course outcomes. Second, our comparative design is not limited by variation in methodological approaches, enabling us to more convincingly learn about generalisable mechanisms based on similarities and differences in the country-specific findings. Our selection of countries is based on differences in pre-pandemic social inequalities and differences in fertility trends, welfare regimes, pandemic mitigation policy, pandemic experiences, and data availability. For Denmark, Sweden, and South Australia, we find little to no evidence for pandemic-induced changes in the parental socioeconomic composition of live births. For seven out of nine included European countries (Austria, England, Finland, Netherlands, Scotland, Spain, Wales) and the United States, we find that the COVID-19 pandemic produced a more socioeconomically advantaged birth cohort than expected had pre-pandemic trends continued. For all Latin American countries (Brazil, Colombia, Ecuador, Mexico) included in our analysis, we find the opposite: the composition of live births conceived during the pandemic changed towards more socioeconomically disadvantaged groups. The percentage point differences in the proportion of live births born to advantaged groups were between -1% and +1%, except for Spain (+2.5%) and Ecuador (-3%). Similarly, the percentage point differences in the proportion of babies born to disadvantaged groups were between -1% and +1%, except for Colombia

(+2.6%). We conclude that these changes in socioeconomic composition may cause between-cohort differences in life course outcomes that are affected by the socioeconomic position of parents even if in-utero or early life exposure to the pandemic had no direct effect on these outcomes. Cross-country similarities and differences in our results suggest that changes in a birth cohort's socioeconomic composition in response to macro-level shocks depend on both policy responses and on how socioeconomic position relates to agency in fertility decisions in different societies." Page 5, lines 1-28

Discussion

Strengths

"The main strength of our study is its comparative design because it allows us to target a specific statistical estimand and use the same methodological approach to estimate it across 15 countries by using birth register data with almost complete coverage of births. In addition to providing comparable evidence for pandemic-induced compositional change across countries, this study design enables us to present a powerful empirical foundation for generalisable population-level mechanisms based on similarities and differences in results across countries. Although the effect size of compositional changes varied across countries, the consistency in similarities between European countries and the United States and in the differences between these countries and the Latin American countries support our proposed mechanisms behind macro-level shocks and corresponding changes in birth cohort composition. Further, changes in socioeconomic composition were visible irrespective of whether we used maternal or parental education, household income, or area-level deprivation of the mother's residential area as an indicator of socioeconomic circumstances. The deviation of results for the exceptional pandemic cases Sweden and South Australia gives additional credibility to our results and their interpretation." Page 24, lines 9-22

We believe it is helpful to describe the process of this study to justify it despite the now published country-specific analyses. When we formulated the Lockdown Cohort-effect hypothesis and started working on this in 2022, many of these papers did not exist and only a few were available as preprints. The first results based on population-wide data from Scotland (a published commentary), Spain (preprint), and later Norway (preprint) and results based on survey data on fertility intentions during the pandemic supported our initial hypothesis. However, the problem was that these separate studies used different methodological approaches. While using different study designs to answer the same research question with the same data can strengthen the robustness of the results, using different methods for different data and slightly different research questions (estimands) leaves us wondering if the results were the same, had everyone targeted the same estimand with the same method using different data.

Our comparative approach, although it prevented us from publishing earlier, allows us to learn more convincingly about macro-level mechanisms because we can target one specific research question and use the same study design across countries to answer it. The alternative, starting a narrative review of the existing literature now, would never reach the same level of explanatory power because comparability of results would be severely limited by the differences in what exactly country-specific analyses estimated and how they estimated it. While country-specific analysis can answer if compositional changes happened in that country, their results can hardly be used to describe more general macro-level mechanisms of how birth cohort composition changed in reaction to the COVID-19 pandemic and thus how we may expect birth cohort changes during future macro-level shocks.

Additionally, we want to defend the central position of our motivation regarding potentially biased conclusions of existing and future studies that do not consider compositional change in observed (and unobserved) parental characteristics. While there are now several papers providing country-specific evidence for compositional change, there are even more papers estimating the effect of the COVID-19 pandemic on early life outcomes (birth outcomes) as mentioned in the introduction. Studies on the former topic are mostly carried out by demographers whose main interest is in fertility. Their evidence on compositional change is a byproduct of their interest in heterogeneous fertility responses to the pandemic and (apart from now three exceptions we are aware of) these papers do not discuss the potential effects these compositional changes can cause for other outcomes, for example preterm birth or future life course outcomes of this cohort. On the other hand, studies on the latter topic were (until now) mostly carried out by medical researchers interested in perinatal outcomes. These studies, conversely, do not consider pandemic-induced compositional changes that could bias their conclusions regarding improved outcomes (for example, preterm birth). Here, consideration of selection into conception would be relevant for studies including babies born 22 weeks after the start of the pandemic and later. In this paper, we wanted to bridge these two seemingly disconnected debates by focusing on the consequences of potential compositional changes from the beginning.

3. Comment

"In the discussion the authors write: "Socioeconomic differences in the fertility response to the pandemic have recently been confirmed in [several] country-specific analyses... We add to this literature by focussing on changes in the socioeconomic composition of cohorts conceived and born during the COVID-19 pandemic." But socioeconomic differences in the fertility response and changes in the socioeconomic composition of conceived cohorts are the same thing. "

Response

We thank the reviewer for the close reading of our manuscript.

We agree with the reviewer that this is not distinctive enough and have changed this part in the manuscript – also aligned with changes in response to the second comment. Further, we have updated the discussion of previous literature by papers that have been published after we submitted our manuscript in March 2024.

Comparison with previous literature on socioeconomic differences in fertility during the COVID-19 pandemic

"Socioeconomic differences in the fertility response to the pandemic have recently been confirmed in country-specific analyses of Spain²⁷, Norway²⁸, Iceland⁵², Sweden⁵³, the United States^{34,35}, Colombia³⁶, Brazil³⁶, and Australia⁵⁴. In Brazil and Colombia, fertility was found to have decreased for women with at least 8 years of schooling while this effect was null or positive for women with fewer years of education.³⁶ In Spain²⁷, Norway²⁸, and the United States³⁴, more babies than expected were born to

women with tertiary education, and a decrease (Spain) or little to no change in live births among women with less formal education (Norway, United States). Similarly, in Iceland, the fertility increase in 2021 was mainly driven by the increase of third births - especially among women with tertiary education and high income.⁵² Australia also experienced an overall increase in birth rates but areas with higher unemployment, lower incomes, and a larger share of non-English speaking residents showed a slower growth in birth rates.⁵⁴ For Sweden, Ohlsson-Wijk and Andersson⁵³ found an increase in the relative risk of first and second births among Swedish-born mothers with 'high income' compared to Swedish-born mothers with 'medium income' in 2020. However, a comparison between their and our results are difficult because Ohlsson-Wijk and Andersson only provide age-adjusted results for Swedish-born women and first and second births by different labour market activity while we estimate compositional changes in the entire birth cohort by household income, education, and maternal age.

We add to this demographic evidence by specifically estimating percentage-point changes in the socioeconomic composition of entire birth cohorts conceived and born during the COVID-19 pandemic in a comparable approach across 15 countries. This wider focus is warranted by the under-recognized potential effect of sudden compositional changes of births on population-level differences in outcomes that are associated with socioeconomic circumstances at birth and/or parental socioeconomic position. Moreover, our comparative design enables us to abstract generalisable mechanisms from similarities and differences in the results across countries." Page 19, lines 3-26

4. Comment

"A more direct motivation for the paper could be to start with what is known about compositional changes; what are the remaining open questions; what this paper is doing; and only then discussing the implications for the study of pandemic cohort outcomes."

Response

We thank the reviewer for this suggestion and want to refer to our extensive changes to the introduction as our response to the reviewer's first and second comment.

We believe the suggested new structure of the introduction would be a valid alternative to our approach – especially if the paper was mainly addressing demographers interested in fertility. However, our paper is aimed at a wider audience of health and social scientists that will sooner or later study future life course outcomes of the December 2020-December 2021 cohort – either with the goal to study the long-term effects of pandemics or just because individuals of this cohort are included in their study population. A main goal of our paper is to bring compositional changes during the COVID-19 pandemic and their potential consequences for population-average outcomes to the attention of researchers studying health, developmental, and socioeconomic outcomes as compositional changes have not received the attention they may deserve in these studies (kindly see comment above and revised introduction). We thus wanted to use the introduction to make a strong case that researchers should consider compositional changes resulting from heterogenous fertility responses to the pandemic if individuals from this birth cohort are included in their study.

5. Comment

“ Figures 3 - 7 and Table 1 all show variations of the same information which is a bit confusing because as a reader one expects that different exhibits show different / new information. Also Figure 2 shows patterns that are very similar across all countries and this information can be summarized more efficiently in a sentence or two in the text (and the figure can be moved to the appendix). It is also not clear that Figure 1 is appropriate for the introduction since this paper is not about the effect of the pandemic on life course outcomes.”

Response

We thank the reviewer for this suggestion to reduce the number of figures. We have moved Figure 2 (visualising the stringency index) to the supplement and instead summarised the information in the method section as suggested by the reviewer. We have removed Figure 1 from the introduction. We have also removed figure 6 (ranking effect sizes for compositional change of most advantaged group by country) from the manuscript as it is not showing sufficiently different information than figure 7.

However, we believe Figures 3, 4, 5, and 7 and Table 1 are important to keep in the main manuscript. (Previous) Figure 3 presents the simple description of proportions of births in the most and least advantaged groups. This gives the reader a good overview of the data and changes that can be seen without any modelling. Figure 4 and 5 are important because i) they make clear that we are modelling counts (instead of the proportions presented in Figure 3), ii) they show our analytical approach intuitively and allow the reader to judge if our simple model specification fits the data sufficiently, and iii) because they give the reader a sense of the changes in the number of births that underlie the proportions as an increase in the share of births held by the most advantaged groups does not necessarily imply an increase in the number of births in that group. We also want to keep Table 1 because it shows the most relevant numbers for all countries in one place.

To guide the reader through the results section, we now include this explanation briefly at the start of the results section as the methods section only appears after the discussion section:

Results

“The analysed data covered 77.95 million live births across 15 countries born between 2015-2021 out of which over 10.9 million live births were conceived during the COVID-19 pandemic. We first give a descriptive overview of trends in the proportion of live births born to the socioeconomically most disadvantaged and advantaged groups in each country in Figure 1. Next, we show the estimated observed and counterfactual trends in the number of births in these groups in Figure 2 and Figure 3 to present our analytical approach and the changes in the number of births that underlie the estimated counterfactual socioeconomic composition of birth cohorts. Finally, we present our main results, the percentage-point differences between the observed and the counterfactual socioeconomic compositions, in Figure 4. The most important results visualised in Figures 1 to 4 are then summarised in Table 1.” Page 6, lines 2-11

Reviewer #2:

“The authors of this article study how the COVID-19 pandemic influenced the socioeconomic composition of newborns. They draw from an exceptional amount of high-quality data from 12 high- and middle-income countries. They highlight the vast and heterogeneous impact of the pandemic on the cohort of newborns and draw implications for future studies, offering a cautionary tale for researchers examining this cohort of children.

The study is well-executed and provides a comprehensive and valuable effort to understand the consequences of the pandemic. The methodology is sound and well explained. The results align with other studies showing how patterns of selection into conception and child outcomes differed across contexts, such as in Spain and Chile. I believe the authors have done exceptional work, and I have no further comments. This study offers a substantial contribution to our understanding of the consequences of the COVID-19 pandemic.”

Response

We are grateful for this positive assessment. We hope that the reviewer agrees with the revised version too.

Reviewer #3:**Comment:**

“The work is of good quality in all its sections: the literature review, the quality of its data sources, the data analysis, and the discussion, which includes discussing the probable mechanisms behind its descriptive results. Above all, it successfully demonstrates what it aims to show: the change in the socioeconomic composition of births conceived during the COVID-19 pandemic. Suggesting different mechanisms for different birth composition seems to me to be one of the strong points of the paper, derived from a solid theoretical assumption: that changes in a birth cohort’s socioeconomic composition in response to macro-level shocks depend on how socioeconomic position relates to agency in fertility decisions.”

Response

We want to thank the reviewer for this positive review and hope that the reviewer agrees with the revisions in response to another reviewer’s comments.

1. Comment

“The solution found to the difficulties posed by the various indicators for parental socio-economic circumstances are also acceptable. However, it would be good to devote a little more space to commenting on the potential problems that the well-known delay in the timely annual registration of

births during the pandemic, at least in some cases, such as Mexico, may cause. More space could also be devoted to justifying the choice of countries in the sample.”

Response

Thank you for this suggestion. We have given more space to the issue of late registration (which was previously more prominently discussed in the country profiles) in the main text. We now also make a stronger case for our comparative design (compared to country-specific analyses) which emphasises that carrying out a unified analysis across countries enables us to propose generalisable mechanisms. For this, it was important to include countries with different pre-existing welfare and fertility regimes and pandemic experiences and policy responses. During the revisions we have added data from Sweden and South Australia as interesting exceptions regarding pandemic experience and Denmark, which we would expect to behave more like Finland, the Netherlands, and Austria based on their similarities in welfare regime. However, it is also clear that the choice of included countries is limited by data availability. We mention this in the revised introduction when describing the study’s aims.

Introduction

“In this study, we use population-wide administrative data on 77.9 million live births from 15 countries covering the period 2015-2021 to analyse changes in the parental socioeconomic composition of babies conceived during the COVID-19 pandemic (born between December 2020 and December 2021). The contribution of this comparative study is twofold. First, we bridge disconnected debates in demography on fertility responses to the pandemic and in epidemiology on the pandemic’s effect on pregnancy and birth outcomes by providing comparable evidence on compositional changes that can cause between-cohort differences in life course outcomes. Second, our comparative design is not limited by variation in methodological approaches, enabling us to more convincingly learn about generalisable mechanisms based on similarities and differences in the country-specific findings. Our selection of countries is based on differences in pre-pandemic social inequalities and differences in fertility trends, welfare regimes, pandemic mitigation policy, pandemic experiences, and data availability. For Denmark, Sweden, and South Australia, we find little to no evidence for pandemic-induced changes in the parental socioeconomic composition of live births. For seven out of nine included European countries (Austria, England, Finland, Netherlands, Scotland, Spain, Wales) and the United States, we find that the COVID-19 pandemic produced a more socioeconomically advantaged birth cohort than expected had pre-pandemic trends continued. For all Latin American countries (Brazil, Colombia, Ecuador, Mexico) included in our analysis, we find the opposite: the composition of live births conceived during the pandemic changed towards more socioeconomically disadvantaged groups. The percentage point differences in the proportion of live births born to advantaged groups were between -1% and +1%, except for Spain (+2.5%) and Ecuador (-3%). Similarly, the percentage point differences in the proportion of babies born to disadvantaged groups were between -1% and +1%, except for Colombia (+2.6%). We conclude that these changes in socioeconomic composition may cause between-cohort differences in life course outcomes that are affected by the socioeconomic position of parents even if in-utero or early life exposure to the pandemic had no direct effect on these outcomes. Cross-country similarities and differences in our results suggest that changes in a birth cohort’s socioeconomic composition in response to macro-level shocks depend on both policy responses and on how socioeconomic position relates to agency in fertility decisions in different societies.” Page 5, lines 1-28

Limitations

“In Austria, Denmark, Colombia, Mexico, the Netherlands, and Spain the differences between the observed and counterfactual number of live births without information on socioeconomic circumstances were non-negligible when compared to differences in the number of births in the other population groups. Thus, increases or decreases in the number of births for population groups could be affected by missing data or misclassification of parental socioeconomic position. Another alternative explanation for observed compositional changes, among included countries (especially Ecuador and Mexico) where delayed registration of birth is common, could be that the registration of births was drastically delayed for some population groups during the COVID-19 pandemic. However, if delayed registration of births was more common among disadvantaged population groups in these countries, the inclusion of these births would lead to an even stronger estimated compositional shift towards a socioeconomically more disadvantaged birth cohort than our estimates show. Where relevant, the issue of delayed birth registration is discussed in more detail in the country profiles (see suppl. material).” Page 25, lines 1-13

2. Comment

“The solution to another common problem, generating a counterfactual scenario of expected births in the absence of a pandemic, is also well solved in the data analysis, although the paper will benefit from stating more clearly why assuming that “pre-pandemic trends continued instead” is the best option (in the context of fertility decline, when fertility reaches very low levels, a model could also indicate that the decline was going to stop or even ‘rebound’)”

Response

Thank you for this suggestion. As in response to the editor’s similar comment, we now provide a clearer justification of the choice of our counterfactual assumption in the method section. Moreover, we now discuss alternative strategies at the beginning of the limitation section.

Analytical strategy

“We used the parameters estimated by the models described above to estimate the sum of the group-specific number of births over the exposed period (December 2020 – December 2021) had the COVID-19 pandemic not happened and, counter to the fact, pre-pandemic trends continued instead.⁵⁰ This estimation of the counterfactual is justified as we only estimate the number of births for a short period of 13 months for monthly data or 54 weeks for weekly data. Assuming abrupt changes in the level, secular trend, and seasonality of the number of births between December 2020 and December 2021 in absence of the pandemic would be less realistic.” Page 31, lines 3-9

Limitations

“In our counterfactual scenario, we assumed that pre-pandemic trends in the population-group-specific number of live births had continued in absence of the COVID-19 pandemic. Alternative counterfactual scenarios could have assumed abrupt changes in the level of the number of births, a sudden reversal in secular trends (for example, an increase after a long decline) in the number of births, and/or changes in the seasonality of birth counts for some or all population groups during the 13-month period between December 2020 and December 2021. However, in our view, these counterfactuals would require stronger justification than our approach and weaken comparability across countries.” Page 24, lines 24-31

3. Comment

“Recent Latin American and European scientific literature on the topic are well taken into account. Finally, although the paper might not have space for great theoretical disquisitions, it would be good to mention that reproductive behaviour has been studied and modelled from various perspectives that could add theoretical thickness to studies on the subject, such as the Theory of Planned Behaviour or others.”

Response

Thank you for referring us to the Theory of Planned Behaviour. We have now included a new paragraph in our discussion of previous fertility literature that refers to the Theory of Planned Behaviour.

Discussion

Comparison with previous literature on socioeconomic differences in fertility during the COVID-19 pandemic

“Changes in reproductive behaviours during the COVID-19 pandemic could be explained by using various theoretical perspectives. For example, seen through the lens of Planned Behaviour Theory³⁶, the pandemic may have changed the intention to have a child because of potential negative consequences (e.g., increased health risks for pregnant women and offspring), a perceived increase in ‘interfering’ factors (e.g., economic uncertainty, income losses, or restricted access to reproductive technology), or perceived increases in ‘enabling’ factors (e.g., work-life balance, or lower opportunity costs).” Page 20, lines 10-16

We thank the reviewer for their critical reading of our paper and the editor for the invitation to revise our manuscript. Kindly see our responses to the reviewer's comment below.

Reviewer #1

"Thank you for the detailed responses to my comments. I think the paper has become clearer and more concise.

I think it is still not clear what exact questions the paper tries to answer that have not been answered by the existing literature. The authors write that their they "bridge disconnected debates in demography on fertility responses to the pandemic" but what exactly these "disconnected debates" are is unclear. I agree with the editor and the other reviewers that there is scientific value in replicating and expanding existing analyses across a broad set of countries using a uniform methodological framework. But it is still important to clarify the exact priors from the existing literature."

Response:

We have tried to improve the introduction in two ways in response to your comment.

First, we revised the part about "disconnected debates" in the literature. To achieve this, we have also rearranged the later parts of the introduction. We also exemplarily highlight findings from previous literature.

Introduction:

Page 4, line 2-5

We believe that such compositional changes in parental socioeconomic characteristics are especially likely in the context of the unequal impacts of the COVID-19 pandemic.¹⁵⁻¹⁸ There are many plausible hypotheses for why fertility responses to the pandemic might have differed across socioeconomic or age groups¹⁹ and empirical evidence has also shown differing fertility responses²⁰⁻²⁸. For example, a study using Norwegian register-based data has concluded that an increase in fertility during the pandemic was driven by women aged 28-35, women who already have children, and women with university education.²² For Spain, Cozzani et al. have found a pandemic-induced decline in fertility especially among women without university education.²¹

Page 4, line 28 and onwards

"Demographic research focused on the COVID-19 pandemic's effect on fertility^{22,23,26,31,35,36} is not necessarily concerned with the consequences of changed fertility for life course outcomes of babies born and conceived during the pandemic. Conversely, medical research which focuses on the COVID-19 pandemic's effect on perinatal outcomes like preterm birth, birth weight, or stillbirths rarely discusses how selection in-utero and/or selection into conception may affect their conclusions regarding found improved or worsened perinatal outcomes.¹⁰⁻¹² Apart from a few recent exceptions, scientific discussion on the COVID-19 pandemic's effect on life course outcomes seems disconnected from the demographic literature on the pandemic's (heterogenous) effect on fertility. Yet, compositional changes through selection in-utero and/or selective conception could explain pandemic effects on perinatal outcomes. For example, Catalano et al. argue that, for the United States, the decline in preterm births for babies conceived before but born during the pandemic was caused by selection in-utero.³⁷ Similarly,

Cozzani et al. argue that the pandemic's positive effects on preterm births among babies conceived during the pandemic are caused by selective conception in Spain.²¹

Not accounting for compositional changes when studying in-utero exposure to a pandemic on life course outcomes has led to erroneous conclusions in the social and health sciences before. Only in 2022, it has been shown that, in the United States, war-induced changes in socioeconomic composition of parents during the 1918 influenza pandemic explain why the 1919 birth cohort had lower adult socioeconomic status than earlier and later birth cohorts.³⁸ Previously, this difference was thought to be caused by in-utero exposure to the 1918 influenza pandemic.⁶

Thus, if the COVID-19 pandemic has led to sudden changes in the parental socioeconomic composition of the cohort born and conceived during the pandemic, research comparing (persons from) this birth cohort to other cohorts studying trends in life course outcomes needs to take this compositional change into consideration.”

Second, we now take more space to present our precise research question: “*Do the cohorts of live births conceived during the COVID-19 pandemic have a different parental socioeconomic composition than expected had there been no pandemic?*” and then immediately refer to existing literature on heterogenous fertility responses. In 2022, we did not set out to “replicate” existing country-specific but concede that this is what our study, to some extent, does in the now existing literature as the reviewer has pointed out.

We hope that this sufficiently clarifies the position of this paper in the existing literature together with our mentioning of existing fertility studies in our introduction and discussion. Please note that we compare our results with this existing literature and provide a detailed discussion in the revised discussion section.

Page 5, line 17 onwards

“In this study, we analyse population-wide administrative data on 77.9 million live births from 15 countries covering the period 2015-2021 to answer if, and to what extent, the cohorts of live births conceived during the COVID-19 pandemic (born between December 2020 and December 2021) have a different parental socioeconomic composition than expected had there been no pandemic. We thereby replicate and expand existing country-specific studies on socioeconomic inequalities in the pandemic's effect on fertility for 8 of 15 countries (Brazil, Colombia, Mexico, Scotland, South Australia, Spain, Sweden, the United States) and add new evidence for the other 7 included countries (Austria, Denmark, Ecuador, England, Finland, Netherlands, Wales). Apart from this, our comparative study makes two contributions. First, we bridge the literature in demography on heterogenous fertility responses to the pandemic and the literature in epidemiology on the pandemic's effect on perinatal outcomes by providing comparable evidence on compositional changes that can cause between-cohort differences in perinatal outcomes and other life course outcomes. We thereby aim to spread awareness about potential long-term implications of changes in the socioeconomic composition of babies conceived during the pandemic beyond the sphere of fertility research. Second, our comparative design is not limited by variation in the specific research questions and methodological approaches as present across the existing country-specific literature. This enables us to more convincingly learn about generalisable mechanisms based on similarities and differences in the country-specific findings.”

Reviewer #2

"I congratulate the authors as the manuscript improved even more."

Response:

Thank you for this assessment.

Reviewer #3:

"I recommend the publication of the revised version"

Response:

Thank you.